# Chronic postnatal chemogenetic activation of forebrain excitatory neurons evokes persistent changes in mood behavior

Sthitapranjya Pati[1]*, Kamal Saba[2†], Sonali S Salvi[1†], Praachi Tiwari[1], Pratik R Chaudhari[1], Vijaya Verma[3], Sourish Mukhopadhyay[1], Darshana Kapri[1], Shital Suryavanshi[1], James P Clement[3], Anant B Patel[2], Vidita A Vaidya[1]*

[1]Department of Biological Sciences, Tata Institute of Fundamental Research, Mumbai, India; [2]Centre for Cellular and Molecular Biology, Hyderabad, India; [3]Neuroscience Unit, Jawaharlal Nehru Centre for Advanced Scientific Research, Bengaluru, India

**Abstract** Early adversity is a risk factor for the development of adult psychopathology. Common across multiple rodent models of early adversity is increased signaling via forebrain Gq-coupled neurotransmitter receptors. We addressed whether enhanced Gq-mediated signaling in forebrain excitatory neurons during postnatal life can evoke persistent mood-related behavioral changes. Excitatory hM3Dq DREADD-mediated chemogenetic activation of forebrain excitatory neurons during postnatal life (P2–14), but not in juvenile or adult windows, increased anxiety-, despair-, and schizophrenia-like behavior in adulthood. This was accompanied by an enhanced metabolic rate of cortical and hippocampal glutamatergic and GABAergic neurons. Furthermore, we observed reduced activity and plasticity-associated marker expression, and perturbed excitatory/inhibitory currents in the hippocampus. These results indicate that Gq-signaling-mediated activation of forebrain excitatory neurons during the critical postnatal window is sufficient to program altered mood-related behavior, as well as functional changes in forebrain glutamate and GABA systems, recapitulating aspects of the consequences of early adversity.

*For correspondence:
sthita.pati@gmail.com (SP);
vvaidya@tifr.res.in (VAV)

†These authors contributed equally to this work

## Introduction

Early-life experience plays a crucial role in the maturation and fine-tuning of neurocircuitry that drives emotional behavior in adulthood (*Hensch, 2004*; *Hensch, 2005*; *Bale et al., 2010*; *Berardi et al., 2000*; *Carr et al., 2013*; *Kessler et al., 2010*). Both clinical and preclinical evidence indicates that early-life adversity serves as a key risk factor for the development of adult psychopathology, increasing susceptibility to psychiatric disorders like anxiety, major depression and schizophrenia (*Carr et al., 2013*; *Anda et al., 2006*; *Knuesel et al., 2014*; *Wright et al., 1995*; *Glover, 2011*). Stressful experiences in adulthood can produce behavioral alterations that are often transient in nature, however perturbations in the vulnerable perinatal 'critical window' can program lasting changes in emotional behavior (*Heim and Nemeroff, 2001*; *McEwen, 2003*; *Ogle et al., 2015*). Several animal models have been used to study the persistent behavioral changes caused by early-life perturbations, and have been instrumental in understanding specific underlying neural mechanisms involved in the programming of adult emotional behavior (*Glover, 2011*; *Heim and Nemeroff, 2001*; *Francis et al., 1999*; *Walker and McCormick, 2009*; *Weinstock, 2008*; *Welberg and Seckl, 2001*; *Ansorge et al., 2007*; *Ansorge et al., 2004*).

**eLife digest** Stress and adversity in early childhood can have long-lasting effects, predisposing people to mental illness and mood disorders in adult life. The weeks immediately before and after birth are critical for establishing key networks of neurons in the brain. Therefore, any disruption to these neural circuits during this time can be detrimental to emotional development. However, it is still unclear which cellular mechanisms cause these lasting changes in behavior.

Studies in animals suggest that these long-term effects could result from abnormalities in a few signaling pathways in the brain. For example, it has been proposed that overstimulating the cells that activate circuits in the forebrain – also known as excitatory neurons – may contribute to the behavioral changes that persist into adulthood.

To test this theory, Pati et al. used genetic engineering to modulate a signaling pathway in male mice, which is known to stimulate excitatory neurons in the forebrain. The experiments showed that prolonged activation of excitatory neurons in the first two weeks after birth resulted in anxious and despair-like behaviors as the animals aged. The mice also displayed discrepancies in how they responded to certain external sensory information, which is a hallmark of schizophrenia-like behavior. However, engineering the same changes in adolescent and adult mice had no effect on their mood-related behaviors.

This animal study reinforces just how critical the first few weeks of life are for optimal brain development. It provides an insight into a possible mechanism of how disruption during this time could alter emotional behavior. The findings are also relevant to psychiatrists interested in the underlying causes of mental illness after early childhood adversity.

The prenatal period and the first few weeks after birth are marked by the establishment and functional maturation of several neurocircuits in rodent models, representing a critical period in which these circuits are particularly amenable to modification by environmental stimuli (*Hensch, 2005*; *Carr et al., 2013*). Rodent models of early-life perturbations encompass those based on gestational stress (*Glover, 2011*), maternal immune activation (*Knuesel et al., 2014*; *Wright et al., 1995*), disruption of dam-pup interaction (*Liu et al., 1997*; *Levine and Lewis, 1959*) or pharmacological treatments (*Oberlander et al., 2009*; *Cutler et al., 1996*; *Sarkar et al., 2014a*), and exhibit both distinct and overlapping behavioral and physiological effects in adulthood (*Carr et al., 2013*; *Heim and Nemeroff, 2001*). Strikingly, a commonality noted across these animal models is the fact that multiple molecular, cellular, functional and behavioral changes often persist throughout the animal's lifespan (*Carr et al., 2013*; *Heim and Nemeroff, 2001*; *Suri and Vaidya, 2015*). Amongst the underlying mechanisms implicated in the establishment of such long-lasting changes in response to early-life perturbations are a dysregulation of the hormonal stress response pathway (*Oberlander et al., 2009*; *Kalinichev et al., 2002*; *Leussis et al., 2012*; *Wilber et al., 2009*; *Fish et al., 2006*; *Gillespie et al., 2009*), serotonergic system (*Altieri et al., 2015*; *Gross and Hen, 2004*; *Shah et al., 2018*), and emergence of excitation-inhibition balance within key cortical neurocircuits (*Sohal and Rubenstein, 2019*; *Gatto and Broadie, 2010*).

Common across several rodent models of early-life perturbations are alterations in G protein-coupled receptor (GPCR) signaling, including via the serotonin$_{1A}$ (5-HT$_{1A}$) receptor (*Gross et al., 2002*; *Richardson-Jones et al., 2010*; *Goodfellow et al., 2009*), serotonin$_{2A}$ (5-HT$_{2A}$) receptor (*Benekareddy et al., 2010*; *Benekareddy et al., 2011*; *Weisstaub et al., 2006*; *Malkova et al., 2014*; *Wischhof et al., 2015*), metabotropic glutamate receptors 1 and 5 (mGluR1/5) (*Genty et al., 2018*; *Lin et al., 2018*), muscarinic acetylcholine receptor 1 (M1) (*Proulx et al., 2014*) and $\alpha_1$ adrenergic receptors (*Loria and Osborn, 2017*). The emergence of aberrant emotional behavior in these animal models has been suggested to involve a key role for both excitatory Gq-coupled and inhibitory Gi-coupled GPCRs, in particular an appropriate balance of signaling between the Gq-coupled 5-HT$_{2A}$ receptor and the Gi-coupled 5-HT$_{1A}$ receptor in the forebrain has been hypothesized to be a critical determinant of the establishment of emotional behavior (*Sarkar et al., 2014b*; *Sargin et al., 2019*; *Lambe et al., 2011*; *Vinkers et al., 2010*). Enhanced signaling via the cortical 5-HT$_{2A}$ receptor is thought to be one of the common features noted across distinct models of early-life perturbations, including maternal separation (*Benekareddy et al., 2010*; *Benekareddy et al., 2011*), postnatal

fluoxetine (*Sarkar et al., 2014b*) and maternal immune activation (*Malkova et al., 2014*; *Moreno et al., 2011*). Interestingly, a systemic blockade of the Gq-coupled 5-HT$_{2A}$ receptor overlapping with early stress or postnatal fluoxetine treatment can prevent the emergence of adult anxiety and depressive behavior, and associated molecular and cellular correlates (*Benekareddy et al., 2011*; *Sarkar et al., 2014b*). Furthermore, pharmacological stimulation of the 5-HT$_{2A}$ receptor during the postnatal critical window is sufficient to evoke a persistent increase in anxiety in adulthood (*Sarkar et al., 2014b*). Collectively, these observations motivate the key question of whether perturbed Gq-coupled signaling within the forebrain in the critical postnatal window plays an important role in the establishment of persistent changes in mood-related behaviors.

Here, we have tested the hypothesis that enhanced Gq-mediated signaling in forebrain excitatory neurons during the postnatal critical window may be sufficient to program persistent alterations in mood-related behavior in adulthood. To address this central question we expressed the excitatory Designer Receptors Exclusively Activated by Designer Drugs (DREADD) in CamKIIα-positive forebrain excitatory neurons using a bigenic mouse line (CamKIIα-tTA::TetO hM3Dq) (*Alexander et al., 2009*), and chemogenetically activated Gq signaling through oral administration of the DREADD agonist clozapine-N-oxide (CNO; 1 mg/kg) from postnatal Day 2 to 14 prior to behavioral analysis in adulthood. Our findings demonstrate that chemogenetic activation of Gq signaling in CamKIIα-positive forebrain excitatory neurons by chronic postnatal CNO (PNCNO) treatment enhances anxiety- and despair-like behavior, accompanied by impaired sensorimotor gating in adulthood. These long-lasting behavioral changes evoked by PNCNO treatment are associated with a persistent dysregulation of cortical and hippocampal glutamate/GABA metabolism, and perturbed hippocampal excitatory and inhibitory neurotransmission. The criticality of the postnatal time window is highlighted by our observation that the same perturbation performed in the juvenile window or in adulthood has no effect on mood-related behavior. Our findings provide evidence in support of the hypothesis that enhanced Gq signaling within forebrain excitatory neurons during the critical postnatal window is sufficient to evoke perturbed mood-related behavior in adulthood, recapitulating the enhanced vulnerability to psychopathology associated with early adversity.

## Results

### Selective expression and activation of hM3Dq DREADD in CamKIIα-positive forebrain excitatory neurons in CamKIIα-tTA::TetO-hM3Dq bigenic mice during the postnatal window

To examine the persistent behavioral, metabolic, molecular and electrophysiological consequences of postnatal chemogenetic hM3Dq DREADD activation of forebrain excitatory neurons, CamKIIα-tTA::TetO-hM3Dq bigenic mice were generated (*Figure 1A*). This bigenic mouse line is reported to exhibit selective expression of the hM3Dq DREADD in Ca$^{2+}$/calmodulin-dependent protein kinase α (CamKIIα)-positive excitatory neurons in the forebrain (*Alexander et al., 2009*; *Figure 1B*). Western blotting and immunofluorescence analysis confirmed the presence of the HA-tagged hM3Dq DREADD in both the hippocampus and cortex of bigenic mouse pups (P7) (*Figure 1C,D*). Expression of the HA-tagged hM3Dq DREADD was not observed in either the hippocampal subfields or cortex of single-positive, genotype-control mouse pups (P7) (*Figure 1D*). Further, in order to delineate cell type specificity for the expression of the HA-tagged hM3Dq DREADD, we performed double immunofluorescence staining for the HA-tag with the excitatory neuron marker, CamKIIα, the inhibitory neuron marker, GABA and the astrocyte marker, glial fibrillary acidic protein (GFAP) (*Figure 1—figure supplement 1*). We noted that the HA-tagged hM3Dq DREADD exhibited robust co-localization with CamKIIα-positive neurons in the hippocampus and neocortex (*Figure 1—figure supplement 1A,B*). We did not observe any co-localization of the HA-tagged hM3Dq DREADD with either inhibitory neuron marker, GABA or the astrocyte marker GFAP in any of the brain regions examined (*Figure 1—figure supplement 1C*). Further, we noted that the HA-tagged hM3Dq DREADD expression was restricted to the forebrain regions, and we did not observe any HA immunofluorescence in subcortical brain regions including the hypothalamus, pallidum, and periaqueductal gray (*Figure 1—figure supplement 1D*). Collectively, these results indicate the restricted cell type specific expression of the HA-tagged hM3Dq DREADD in forebrain CamKIIα-positive neurons of CamKIIα-tTA::TetO-hM3Dq bigenic mice.

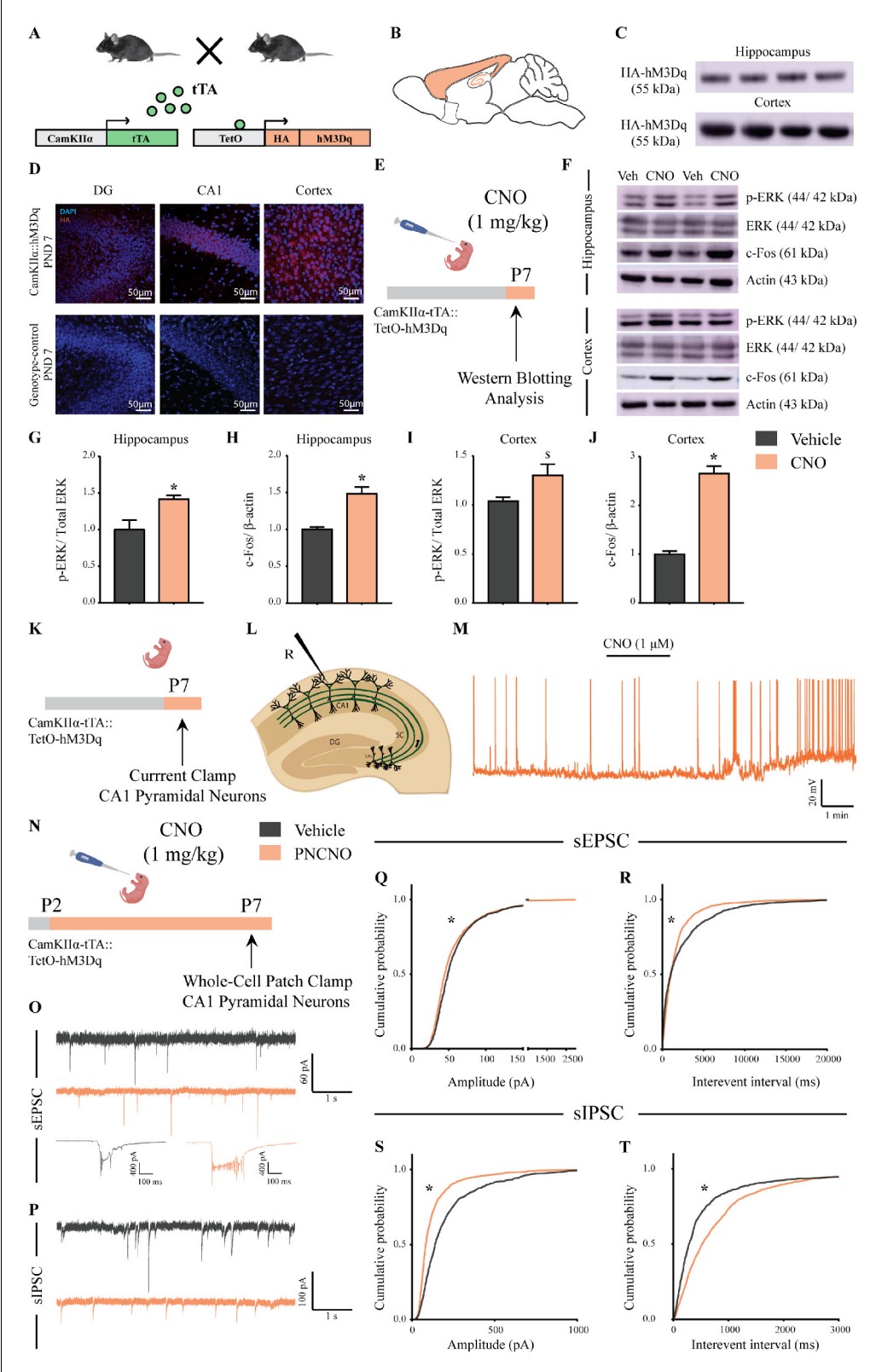

**Figure 1.** Selective expression and activation of hM3Dq DREADD in CamKIIα-positive forebrain excitatory neurons in CamKIIα-tTA::TetO-hM3Dq bigenic mice during the postnatal window. (**A**) Shown is a schematic of the experimental strategy for the generation of the bigenic CamKIIα-tTA::TetO-hM3Dq mouse line to selectively drive the expression of the hM3Dq DREADD in CamKIIα-positive forebrain excitatory neurons. tTA – tetracycline transactivator. (**B**) Shown is a schematic sagittal view of the mouse brain indicating the region of hM3Dq DREADD expression. (**C**) Western blots

*Figure 1 continued on next page*

*Figure 1 continued*

indicate expression of the HA-tag in the hippocampus and the cortex confirming the presence of HA-tagged hM3Dq DREADD (n = 4). (**D**) Shown are representative confocal images indicating expression of hM3Dq DREADD in the DG, CA1, and cortex as identified by HA immunofluorescence, which was not observed in the genotype-control mice (n = 3 per group). (**E**) Shown is the experimental paradigm to assess activity-related signaling signatures following acute CNO-mediated hM3Dq DREADD activation of CamKIIα-positive forebrain excitatory neurons at P7. The mice were fed a single dose of either CNO (1 mg/kg) or vehicle and sacrificed 15 min later for western blotting analysis (n = 4 per group). (**F**) Representative western blots indicate the expression of the neuronal activity-related proteins, p-ERK and c-Fos in the hippocampus and cortex of CNO and vehicle-treated CamKIIα-tTA::TetO-hM3Dq bigenic mouse pups. Densitometric quantification revealed a significant CNO-mediated, hM3Dq DREADD activation evoked increase in p-ERK/ERK (**G**) and c-Fos (**H**) expression in the hippocampi of CNO-treated pups as compared to the vehicle-treated controls (n = 4 per group). In the cortex, hM3Dq DREADD activation resulted in a trend toward an increase in p-ERK/ERK (**I**) and a significant increase in c-Fos (**J**) protein levels in the CNO-treated pups. Results are expressed as the mean ± S.E.M. *p<0.05, $p=0.07 as compared to vehicle-treated controls using the two-tailed, unpaired Student's *t*-test. (**K–L**) Shown is a schematic of the experimental paradigm for whole-cell patch clamp recording from the somata of CA1 pyramidal neurons at P7 in acute hippocampal slices derived from drug-naïve, bigenic CamKIIα-tTA::TetO-hM3Dq mouse pups. R – Recording electrode. (**M**) Bath application of CNO (1 μM) to acute hippocampal slices resulted in hM3Dq DREADD activation mediated robust spiking activity of CA1 pyramidal neurons (n = 3 cells). (**N**) Experimental paradigm to assess the effects of chronic CNO-mediated hM3Dq DREADD activation of CamKIIα-positive forebrain excitatory neurons using whole-cell patch clamp recording. CamKIIα-tTA::TetO-hM3Dq bigenic mouse pups were fed either CNO (1 mg/kg) or vehicle from P2 to P7 followed by recording of sEPSCs and sIPSCs. (**O**) Shown are representative sEPSC traces of vehicle and PNCNO-treated mice at P7. Top traces: examples of small amplitude events. Bottom traces: examples of large-amplitude events. (**P**) Shown are representative sIPSC traces of vehicle and PNCNO-treated mice at P7. (**Q**) PNCNO-treated mice showed significantly altered cumulative probability of sEPSC amplitude with a small decrease at lower amplitudes (<100 pA) and a significant increase in large-amplitude events characterized by a long-tail as compared to vehicle-treated controls. (**R**) PNCNO-treated mice showed a significant decline in the cumulative probability of sEPSC interevent intervals as compared to vehicle-treated controls (n = 7 cells for vehicle; n = 10 cells for PNCNO). PNCNO-treated mice showed a significant decrease in sIPSC amplitude (**S**), and a concomitant increase in sIPSC interevent intervals (**T**) as compared to vehicle-treated controls (n = 6 cells for vehicle; n = 8 cells for PNCNO). Results are expressed as cumulative probabilities. *p<0.001 as compared to PNCNO-treated group using Kolmogorov-Smirnov two-sample comparison. The online version of this article includes the following figure supplement(s) for figure 1:

**Figure supplement 1.** Selective expression of the HA-tagged hM3Dq DREADD in CamKIIα-positive forebrain excitatory neurons in adult CamKIIα-tTA:: TetO-hM3Dq bigenic mice.
**Figure supplement 2.** Enhanced p-ERK/ERK expression following chronic postnatal hM3Dq DREADD activation in CamKIIα-positive forebrain excitatory neurons.
**Figure supplement 3.** Spontaneous network activity and intrinsic excitability following chronic postnatal hM3Dq DREADD activation in CamKIIα-positive forebrain excitatory neurons.
**Figure supplement 4.** Distribution of spontaneous network events following chronic postnatal hM3Dq DREADD activation in CamKIIα-positive forebrain excitatory neurons.

We next assessed whether, acute stimulation of the hM3Dq DREADD by the exogenous ligand, CNO at postnatal Day 7 (P7), resulted in enhanced neuronal activation within the forebrain, using two distinct strategies. First, we performed western blotting analysis to determine the expression levels of the neuronal activity markers, c-Fos and phospho-ERK, following a single dose of CNO (1 mg/kg) administered via feeding to bigenic mouse pups at P7 (*Figure 1E,F*). Western blotting analysis revealed a significant increase in both p-ERK/ERK (*Figure 1G*, $F_{1,6}$ = 5.872, p=0.02) and c-Fos (*Figure 1H*, $F_{1,6}$ = 7.48, p=0.002) levels in the hippocampus of CNO-treated bigenic mouse pups. We also observed a trend toward an increase in p-ERK/ERK levels (*Figure 1I*, $F_{1,6}$ = 8.462, p=0.07) and significant increase in c-Fos levels (*Figure 1J*, $F_{1,6}$ = 5.608, p<0.0001) in the cortex of CNO-treated bigenic mouse pups. Second, we carried out whole-cell patch clamp recordings in current clamp mode from the somata of CA1 pyramidal neurons in acute hippocampal slices derived from bigenic mouse pups (P7) following CNO bath application (*Figure 1K,L*). Bath application of 1 μM CNO evoked robust spiking activity in CA1 pyramidal neurons (*Figure 1M*). These two approaches confirmed that as anticipated, acute postnatal hM3Dq DREADD activation of forebrain excitatory neurons resulted in enhanced neuronal activation.

Given our treatment paradigm involved chronic administration of CNO to mouse pups during the early postnatal window (P2–P14), we further sought to understand the effects of chronic CNO-mediated hM3Dq DREADD activation in CamKIIα-positive forebrain excitatory neurons on neuronal activity, at an interim time point in the midst of chronic CNO administration (P7). Bigenic mouse pups were orally administered CNO (1 mg/kg; PNCNO) from P2 to P7, and then assessed via western blotting analysis for cortical and hippocampal levels of the neuronal activity marker, p-ERK (*Figure 1—figure supplement 2A*), or through electrophysiological measurements of spontaneous excitatory/inhibitory currents in the hippocampus (*Figure 1N*; *Figure 1—figure supplements 3A*

and *4A*). Electrophysiological recordings following this treatment paradigm were carried out in aCSF in the absence of CNO in the bath. We detected a significant increase in p-ERK/ERK levels in the hippocampus (*Figure 1—figure supplement 2B,C*, $F_{1,6}$ = 1.557, p=0.04) and the cortex (*Figure 1—figure supplement 2B,D*, $F_{1,6}$ = 1.069, p=0.0001) of PNCNO-treated mouse pups. Whole-cell patch clamp recording to measure sPSCs and intrinsic membrane properties in CA1 pyramidal neurons from acute hippocampal slices revealed a significant difference in sPSC amplitude in the PNCNO-treatment group, with a small but significant decrease in low amplitude events (<100 pA; *Figure 1—figure supplement 3B,C*, p<0.0001), accompanied by a significant increase in large-amplitude events characterized by the presence of a long-tail in sPSC amplitude event distribution (*Figure 1—figure supplements 3C*, *4B and C*). We also observed a significant reduction in the cumulative probability of sPSC interevent intervals in CA1 pyramidal neurons from the PNCNO- treatment group (*Figure 1—figure supplement 3D*, p<0.0001). CA1 pyramidal neurons in PNCNO-treated hippocampal slices displayed large network activity, characterized by compound negative peaks (*Figure 1—figure supplement 4B*). Out of the total number of events analyzed, the frequency of events greater than 100 pA were almost double in PNCNO-treated CA1 neurons (4.85%) as compared to controls (2.57%). We also noted a small fraction of events (0.8%) with amplitudes greater than 250 pA in CA1 pyramidal neurons from PNCNO-treated mouse pups, which were not detected in vehicle-treated controls (*Figure 1—figure supplement 4D*). In order to understand the influence of CNO-mediated hM3Dq DREADD activation of CamKIIα-positive excitatory neurons during the postnatal window on intrinsic excitability, we plotted an input-output curve by injecting increasing step currents and measured the number of action potentials (*Figure 1—figure supplement 3E*). We observed no change in the number of action potentials generated in CA1 pyramidal neurons of PNCNO-treated mouse pups (*Figure 1—figure supplement 3F*). Measurements of key intrinsic membrane properties revealed no change in input resistance ($R_N$), membrane time constant (τ), sag voltage, and accommodation index in CA1 pyramidal neurons of PNCNO-treated mouse pups (*Table 1*). We did note a trend toward a depolarizing shift in the resting membrane potential (RMP) in CA1 pyramidal neurons of the PNCNO-treatment group as compared to their vehicle-treated controls (*Table 1*, $F_{1,13}$ = 1.862, p=0.06).

We next sought to parcellate the influence of chronic CNO-mediated hM3Dq DREADD activation of CamKIIα-positive forebrain excitatory neurons in the postnatal window on excitatory and inhibitory neurotransmission. Whole-cell patch clamp analysis was carried out to measure sEPSCs and sIPSCs in CA1 pyramidal neurons in acute hippocampal slices derived from bigenic mouse pups treated with CNO (1 mg/kg) or vehicle (*Figure 1N*). We observed a significant difference in sEPSC amplitude in CA1 pyramidal neurons of PNCNO-treated mouse pups as compared to vehicle-treated controls, as revealed by a small but significant decrease in low amplitude events (<100 pA), and a significant increase in large-amplitude events characterized by the presence of a long-tail in sEPSC amplitude event cumulative distribution (*Figure 1O,Q*, p<0.0001). CA1 pyramidal neurons in hippocampal slices from PNCNO-treated mouse pups displayed large sEPSC events characterized by compound negative peaks as compared to vehicle-treated controls (*Figure 1O*; bottom traces). We also noted a significant decline in the cumulative probability of sEPSC interevent intervals in CA1

**Table 1.** Effects of chronic postnatal hM3Dq DREADD activation in CamKIIα-positive forebrain excitatory neurons on intrinsic membrane properties.

| Postnatal Day 7 | CA1 pyramidal neurons | | | | | |
|---|---|---|---|---|---|---|
| Intrinsic Properties | | | | | | |
| Group | RMP (mV) | Input resistance (MΩ) | τ (ms) | Sag (mV) | Sag (%) | Accomodation index |
| Vehicle | −55.96 ± 1.923 | 310.2 ± 39.43 | 34.77 ± 4.62 | −10.09 ± 2.65 | 10.15 ± 2.42 | 0.32 ± 0.06 |
| PNCNO | −51.66 ± 1.151[$] | 352.3 ± 35.23 | 33.62 ± 3.65 | −8.99 ± 1.66 | 9.10 ± 1.22 | 0.48 ± 0.09 |

CamKIIα-tTA::TetO-hM3Dq bigenic mouse pups were fed either CNO (1 mg/kg) or vehicle from P2 to P7, and whole-cell patch clamp was performed at P7 to determine intrinsic membrane properties. No significant effect was noted for input resistance, membrane time constant (τ), sag voltage, percent sag and accommodation index across treatment groups. We noted a trend toward an increase in resting membrane potential (RMP) in hippocampi derived from the PNCNO-treated mouse pups as compared to vehicle-treated controls. [$]p=0.06 as compared to vehicle-treated controls (n = 6 cells for vehicle; n = 9 cells for PNCNO) using the two-tailed, unpaired Student's *t*-test.

pyramidal neurons from the PNCNO-treatment group (*Figure 1R*, p<0.0001). Further, we observed a significant reduction in the cumulative probability of sIPSC amplitude (*Figure 1P,S*, p<0.0001) and an increase in the cumulative probability of sIPSC interevent intervals (*Figure 1T*, p<0.0001) in CA1 pyramidal neurons from PNCNO-treated mouse pups.

Our findings demonstrate the selective expression of the hM3Dq DREADD in the forebrain of CamKIIα-tTA::TetO-hM3Dq bigenic mouse pups, and indicate that acute CNO treatment during postnatal life increases neuronal activity in the hippocampus and cortex. Further, as the main treatment paradigm used in our study is based on chronic CNO-mediated hM3Dq DREADD activation of CamKIIα-positive forebrain excitatory neurons during the postnatal window, our experiments at an interim juncture during postnatal treatment reveal significant changes in both neuronal activity marker expression and electrophysiological measures. Collectively, our results demonstrate that chronic CNO-mediated hM3Dq DREADD activation during the postnatal window results in enhanced expression levels of neuronal activity markers, elevated spontaneous network activity, an increase in spontaneous excitatory currents, and a concomitant decrease in spontaneous inhibitory currents in CA1 pyramidal neurons of the PNCNO-treatment group.

We next sought to assess the persistent behavioral consequences of perturbing neuronal activity of CamKIIα-positive forebrain excitatory neurons during the early postnatal window using chronic CNO-mediated hM3Dq DREADD activation. CamKIIα-tTA::TetO-hM3Dq bigenic mouse pups were orally administered the DREADD ligand, CNO (1 mg/kg), or vehicle, once daily from P2 to to P14 (*Figure 2A*). Postnatal treatment with CNO did not alter the body weight, measured across the period of postnatal treatment or in adulthood (*Figure 2—figure supplement 1A–C*). Chronic CNO-mediated hM3Dq DREADD activation in the early postnatal window did not alter the normal trajectory of sensorimotor development, as indicated by no change in the ontogeny of reflex behaviors, namely surface righting and negative geotaxis, in PNCNO-treated mouse pups as compared to their vehicle-treated controls (*Figure 2—figure supplement 2A–C*).

We examined the influence of chronic CNO-mediated hM3Dq DREADD activation of CamKIIα-positive forebrain excitatory neurons during the early postnatal window on long-lasting changes in anxiety- and despair-like behavior. We subjected bigenic adult mice with a history of PNCNO or vehicle treatment to a battery of behavioral tasks, commencing 3-months post cessation of PNCNO treatment. We performed the open field test (OFT), elevated plus maze (EPM) test, and the light-dark (LD) avoidance test to assess anxiety-like behavior, followed by the forced swim test (FST) to assess despair-like behavior in PNCNO-treated adult bigenic male and female mice (*Figure 2A*, *Figure 2—figure supplement 3A*). We noted a significant increase in anxiety-like behavior in adult male mice with a history of PNCNO treatment on the OFT (*Figure 2B*). The PNCNO-treatment group showed a significant decrease in percent distance traveled in the center (*Figure 2C*, $F_{1, 28} = 1.097$, p=0.03), number of entries to the center (*Figure 2E*, $F_{1, 28} = 1.272$, p=0.02), and total distance traveled in the OFT arena (*Figure 2F*, $F_{1, 28} = 1.23$, p=0.003). The percent time spent in the center of the OFT arena was unchanged across treatment groups (*Figure 2D*). We also noted an increase in anxiety-like behavior in the EPM (*Figure 2G*) in the PNCNO-treated adult male mice, with a significant decline in the number of entries to the open arms (*Figure 2J*, $F_{1, 28} = 2.829$, p=0.02) and a trend toward a decrease in the percent time spent in the open arms (*Figure 2I*, $F_{1, 28} = 1.977$, p=0.08). The percent distance traveled in the open arms (*Figure 2H*) and the total distance traversed in the EPM (*Figure 2K*) were unchanged. Behavioral analysis on the LD avoidance test (*Figure 2L*), revealed an anxiogenic effect of PNCNO treatment in bigenic adult male mice, with a significant decrease in the number of entries to the light box (*Figure 2M*, $F_{1, 28} = 1.229$, p=0.04) and a trend toward a decrease in the time spent in the light box (*Figure 2N*, $F_{1, 28} = 1.378$, p=0.07). We then evaluated the influence of chronic CNO-mediated hM3Dq DREADD activation of forebrain excitatory neurons in the early postnatal window on despair-like behavior in adulthood using the FST (*Figure 2O*). We observed increased despair-like behavior in CamKIIα-tTA::TetO-hM3Dq bigenic adult male mice with a history of PNCNO treatment, as noted by a significant increase in the time spent immobile in the FST (*Figure 2P*, $F_{1, 15} = 7.862$, p=0.03, Welch's correction). Taken together, these results indicate that chronic CNO-mediated hM3Dq DREADD activation of CamKIIα-positive forebrain excitatory neurons during the early postnatal window results in long-lasting increases in anxiety- and despair-like behavior in adult male mice.

Following this, we sought to ascertain whether chronic CNO-mediated hM3Dq DREADD activation of CamKIIα-positive forebrain excitatory neurons during the early postnatal window, evokes a

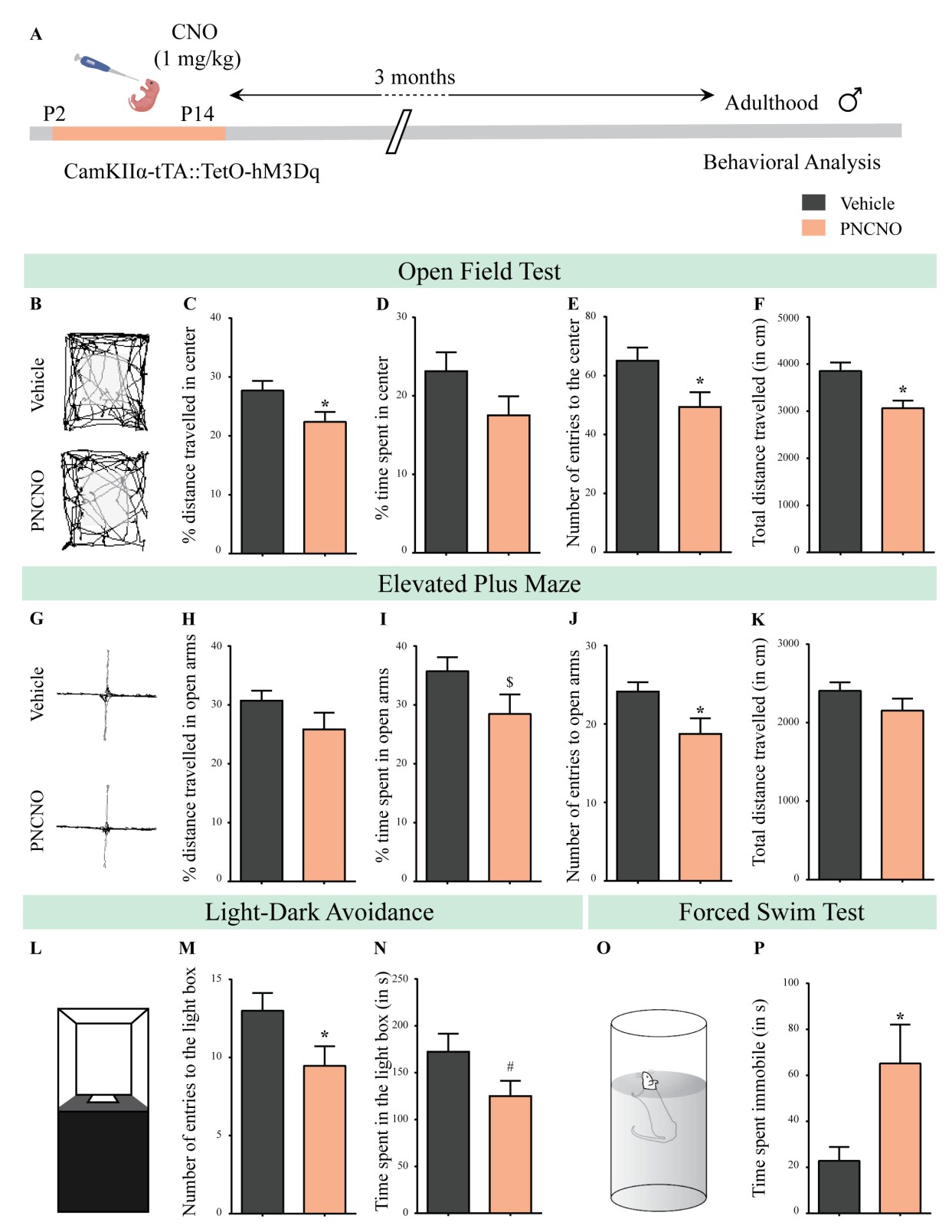

**Figure 2.** Chronic chemogenetic activation of CamKIIα-positive forebrain excitatory neurons during the early postnatal window results in a long-lasting increase in anxiety- and despair-like behavior in adult male mice. (**A**) Shown is a schematic of the experimental paradigm to induce chronic CNO-mediated hM3Dq DREADD activation in CamKIIα-positive forebrain excitatory neurons using bigenic CamKIIα-tTA::TetO-hM3Dq mouse pups that were fed CNO (PNCNO; 1 mg/kg) or vehicle from P2 to P14 and then left undisturbed for 3 months prior to behavioral analysis performed in adulthood on

*Figure 2 continued on next page*

*Figure 2 continued*

male mice. (B) Shown are representative tracks of vehicle or PNCNO-treated CamKIIα-tTA::TetO-hM3Dq bigenic adult male mice in the open field test (OFT). A history of chronic postnatal hM3Dq DREADD activation of CamKIIα-positive forebrain excitatory neurons resulted in increased anxiety-like behavior on the OFT in adulthood, as noted by a significant decrease in the percent distance traveled in center (C), number of entries to the center (E), and the total distance traveled in the OFT arena (F) in PNCNO-treated mice as compared to vehicle-treated controls (n = 15 per group). The percent time spent in the center was not significantly altered (D) in PNCNO-treated mice as compared to vehicle-treated controls. (G) Shown are representative tracks of vehicle or PNCNO-treated CamKIIα-tTA::TetO-hM3Dq bigenic adult mice on the elevated plus maze (EPM). Adult mice with chronic postnatal hM3Dq DREADD activation of CamKIIα-positive forebrain excitatory neurons exhibited increased anxiety-like behavior on the EPM as revealed by a significant decrease in the number of entries to the open arms (J), and a trend toward a decrease in percent time spent in the open arms (I) in PNCNO-treated mice as compared to vehicle-treated controls (n = 15 per group). The percent distance traveled in the open arms (H) and the total distance traveled in the EPM arena (K) was not altered in PNCNO-treated mice as compared to vehicle-treated controls. (L) Shown is a schematic of the light-dark box used to assess anxiety-like behavior. Chronic postnatal hM3Dq DREADD activation of CamKIIα-positive forebrain excitatory neurons resulted in an increased anxiety-like behavior in the LD box test in adulthood, as revealed by a significant decline in the number of entries to the light box (M), and a trend toward decline in the time spent in the light box (N) in PNCNO-treated mice as compared to vehicle-treated controls (n = 15 per group). (O) Shown is a schematic representation of the forced swim test (FST) apparatus used to assess despair-like behavior. Chronic postnatal hM3Dq DREADD activation of CamKIIα-positive forebrain excitatory neurons resulted in an increased despair-like behavior on the FST in adulthood, as revealed by a significant increase in time spent immobile (P) in PNCNO-treated mice as compared to vehicle-treated controls (n = 13 per group). Results are expressed as the mean ± S.E.M. *p<0.05, $p=0.08, #p=0.07; as compared to vehicle-treated controls using the two-tailed, unpaired Student's *t*-test.

The online version of this article includes the following source data and figure supplement(s) for figure 2:

**Source data 1.** Source data for *Figure 2*.

**Figure supplement 1.** Chronic chemogenetic activation of CamKIIα-positive forebrain excitatory neurons during the early postnatal window does not alter weight during CNO administration and in adulthood.

**Figure supplement 2.** Chronic chemogenetic activation of CamKIIα-positive forebrain excitatory neurons during the early postnatal window does not alter the developmental emergence of reflex behaviors.

**Figure supplement 3.** Chronic chemogenetic activation of CamKIIα-positive forebrain excitatory neurons during the early postnatal window results in a long-lasting increase in anxiety-like behavior in adult female mice.

**Figure supplement 4.** Chronic chemogenetic activation of CamKIIα-positive forebrain excitatory neurons during the early postnatal window does not alter repetitive behavior in adult male mice.

**Figure supplement 5.** Chronic CNO administration during the early postnatal window does not influence anxiety- and despair-like behavior in genotype-control, adult male mice.

**Figure supplement 6.** Chronic CNO administration during the early postnatal window does not influence anxiety- and despair-like behavior in C57BL/6J adult male mice.

**Figure supplement 7.** Chronic chemogenetic activation of CamKIIα-positive forebrain excitatory neurons during the early postnatal window using the DREADD agonist Compound 21 (C21) results in a long-lasting increase in anxiety-like behavior in adult male mice.

similar anxiogenic and despair-like behavioral phenotype in adult female mice (*Figure 2—figure supplement 3A*). Bigenic adult female mice with a history of PNCNO treatment exhibited enhanced anxiety-like behavior on the OFT and EPM tests. In the OFT we noted a significant decrease in percent distance traveled in the center (*Figure 2—figure supplement 3B,C*, $F_{1, 20} = 1.438$, p=0.01), and number of entries to the center (*Figure 2—figure supplement 3E*, $F_{1, 20} = 1.158$, p=0.008), with no change observed in other measures (*Figure 2—figure supplement 3D,F*). In the EPM, bigenic adult female mice with a history of PNCNO treatment, showed a significant decrease in the percent distance traveled (*Figure 2—figure supplement 3G,H*, $F_{1, 20} = 3.139$, p=0.04) and the percent time spent in the open arms (*Figure 2—figure supplement 3I*, $F_{1, 20} = 2.31$, p=0.004) as compared to their vehicle-treated controls, with no difference observed on other measures (*Figure 2—figure supplement 3J,K*). PNCNO-treated bigenic adult female mice did not show any change in anxiety-like behavior on the LD avoidance test (*Figure 2—figure supplement 3L–N*). PNCNO-treated bigenic adult female mice did not show any change in despair-like behavior assessed on the FST (*Figure 2—figure supplement 3O,P*). Taken together, these results indicate that chronic CNO-mediated hM3Dq DREADD activation of forebrain excitatory neurons during the early postnatal window results in long-lasting increases in both anxiety- and despair-like behavior in adult male mice, whereas it evokes a persistent increase in anxiety-, but not despair-like behavior, in adult female mice. A caveat to note is that our experiments with adult CamKIIα-tTA::TetO hM3Dq males and females with a history of PNCNO treatment were performed on distinct cohorts at different times. This prevented us from performing a two-way ANOVA analysis to examine sexually dimorphic behavioral effects of excitatory DREADD-mediated chemogenetic activation of forebrain excitatory

neurons during postnatal life. Henceforth, all our studies to assess the behavioral, metabolic, molecular and electrophysiological influence of chronic CNO-mediated hM3Dq DREADD activation of forebrain excitatory neurons during the early postnatal window have been restricted to male mice.

We next sought to determine the influence of chronic CNO-mediated hM3Dq DREADD activation of CamKIIα-positive forebrain excitatory neurons during the early postnatal window on stereotypic behavior. We subjected CamKIIα-tTA::TetO-hM3Dq bigenic adult male mice, with a history of PNCNO treatment to the marble burial test (*Figure 2—figure supplement 4A,B*). We observed no change in stereotypic behavior on the marble burial test, with no difference in the number of marbles buried by the PNCNO or vehicle-treated bigenic adult male mice (*Figure 2—figure supplement 4A–C*). Our observations indicate that chronic CNO-mediated hM3Dq DREADD activation of CamKIIα-positive forebrain excitatory neurons during the early postnatal window does not influence repetitive behavior in CamKIIα-tTA::TetO-hM3Dq bigenic adult male mice.

## Chronic CNO administration during the early postnatal window does not influence anxiety- and despair-like behavior in genotype-control or background strain, adult male mice

Considering the evidence that CNO metabolites can produce off-target behavioral effects (*Gomez et al., 2017*; *MacLaren et al., 2016*), we designed two sets of control experiments which assessed the influence of postnatal CNO administration in genotype-control or background strain mouse pups, and the resultant effects on the programming of adult anxiety- and despair-like behavior. First, we administered CNO (1 mg/kg) or vehicle to genotype-control mouse pups, single- positive for either CamKIIα-tTA or TetO-hM3Dq once daily from P2 to P14 (*Figure 2—figure supplement 5A*). Following a three-month washout period post cessation of CNO treatment, we assayed these mice for anxiety and depressive-like behavior. We did not observe any difference in anxiety-like behavior in the OFT (*Figure 2—figure supplement 5B–E*) in the PNCNO-treated genotype-control cohort as compared to vehicle-treated controls. We did note a small, but significant decrease in total distance traveled in the OFT arena (*Figure 2—figure supplement 5F*, $F_{1, 22}$ = 1.372, p=0.03) in the PNCNO-treated genotype-control group. Behavioral analysis on the EPM indicated no change in anxiety-like behavior in adult genotype-control mice with a history of PNCNO treatment (*Figure 2—figure supplement 5G–K*). In addition, we did not observe any change in anxiety-like behavior in the LD avoidance test (*Figure 2—figure supplement 5L–N*) as a consequence of PNCNO treatment in genotype-control mice. Despair-like behavior was also unchanged across treatment groups, indicating that CNO treatment in genotype-control mice during the postnatal window does not alter behavior on the FST (*Figure 2—figure supplement 5O,P*).

The second control experiment to rule out potential off-target effects of chronic postnatal CNO treatment was performed in the background strain (C57BL/6J). C57BL/6J mouse pups received oral administration of CNO (1 mg/kg) or vehicle once daily from P2 to P14, followed by behavioral testing commencing 3 months post cessation of the CNO treatment regime (*Figure 2—figure supplement 6A*). To assess anxiety-like behavior C57BL/6J adult male mice with a history of PNCNO treatment were tested on the OFT, EPM, and LD avoidance test. We did not observe any change in anxiety-like behavior in the OFT (*Figure 2—figure supplement 6B–F*), EPM (*Figure 2—figure supplement 6G–K*) and the LD avoidance test (*Figure 2—figure supplement 6L–N*) in the PNCNO-treated C57BL/6J adult male mice as compared to their vehicle-treated controls. Despair-like behavior, as assessed by immobility time on the FST was also unchanged across treatment groups, indicating no effect of postnatal CNO treatment in the C57BL/6J background strain (*Figure 2—figure supplement 6O,P*). Collectively, these control experiments indicate postnatal CNO administration does not evoke off-target effects that influence anxiety- and despair-like behavior.

Chronic chemogenetic activation of CamKIIα-positive forebrain excitatory neurons using the hM3Dq DREADD agonist compound 21 (C21) during the early postnatal window results in a long-lasting increase in anxiety-like behavior in adult mice.

We further addressed whether an alternate hM3Dq DREADD agonist compound 21 (C21) when utilized to evoke chronic chemogenetic activation of CamKIIα-positive forebrain excitatory neurons during postnatal life also programs persistent changes in adult anxiety-like behavior. We orally administered C21 to CamKIIα-tTA::TetO-hM3Dq bigenic mouse pups once daily from P2 to 14. We then subjected bigenic adult male mice with a history of PNC21 or vehicle treatment to the open field test (OFT), and elevated plus maze (EPM) test to assess effects on anxiety-like behavior

(*Figure 2—figure supplement 7A*). We noted a significant increase in anxiety-like behavior in adult PNC21 male mice on the OFT (*Figure 2B*). The PNC21 treatment group showed a significant decrease in percent time spent in the center of the OFT arena (*Figure 2—figure supplement 7D*, $F_{1, 28} = 1.252$, p=0.025). The percent distance traveled in the center (*Figure 2—figure supplement 7C*), number of entries to the center (*Figure 2—figure supplement 7E*), and total distance traveled in the OFT arena (*Figure 2—figure supplement 7F*) were unaltered across treatment groups. We also noted an increase in anxiety-like behavior in the EPM (*Figure 2—figure supplement 7G*) in the PNC21-treated adult male mice, with a significant decrease in percent distance traveled in the open arms (*Figure 2—figure supplement 7H*, $F_{1, 28} = 2.295$, p=0.003) and the percent time spent in the open arms (*Figure 2—figure supplement 7I*, $F_{1, 28} = 1.991$, p=0.002). The number of entries to the open arms (*Figure 2—figure supplement 7J*) and the total distance traversed in the EPM (*Figure 2—figure supplement 7K*) were not changed. These results indicate that chronic C21-mediated hM3Dq DREADD activation of CamKIIα-positive forebrain excitatory neurons during the early post-natal window results in long-lasting increases in anxiety-like behavior in adult male mice.

Chronic chemogenetic activation of CamKIIα-positive forebrain excitatory neurons during the juvenile window or in adulthood does not evoke any long-lasting changes in anxiety- and despair-like behavior.

Given we observed that chronic CNO-mediated hM3Dq DREADD activation of CamKIIα-positive forebrain excitatory neurons in the early postnatal window can program persistent changes in anxiety- and despair-like behavior, we next sought to ascertain whether the temporal window in which this perturbation is performed is critical to the establishment of these long-lasting behavioral changes. To address this question, we used chronic CNO-mediated hM3Dq DREADD activation of CamKIIα-positive forebrain excitatory neurons in two distinct temporal windows, namely juvenile life (P28–40) and adulthood. The time duration and dose of CNO treatment was maintained constant across the postnatal, juvenile, and adult treatment paradigms.

CamKIIα-tTA::TetO-hM3Dq bigenic juvenile male mice received CNO (1 mg/kg; JCNO) via oral administration once daily from P28-P40 (*Figure 3A*; *Figure 3—figure supplement 1A*). Following a washout period, we subjected bigenic adult male mice with a history of JCNO treatment to behavioral tests for anxiety- and despair-like behavior. We observed no change in anxiety-like behavior in JCNO-treated mice in the OFT (*Figure 3B*), with no difference noted for the percent distance traveled in the center (*Figure 3C*), percent time spent in the center (*Figure 3D*), number of entries to the center (*Figure 3—figure supplement 1B*) and the total distance traversed in the OFT arena (*Figure 3—figure supplement 1C*). Behavioral testing on the EPM (*Figure 3E*) revealed no influence of JCNO treatment on anxiety-like behavior, with no difference noted for the percent distance traveled in the open arms (*Figure 3F*), percent time spent in the open arms (*Figure 3G*), number of entries to the open arms (*Figure 3—figure supplement 1D*), and total distance traveled in the EPM arena (*Figure 3—figure supplement 1E*). Further, we did not observe any difference in the number entries to the light box (*Figure 3H*) and the time spent in the light box (*Figure 3I*) in JCNO-treated mice on the LD avoidance test. JCNO and vehicle-treated bigenic male mice did not differ on despair-like behavioral measures on the FST, with no significant change in immobility time (*Figure 3J*). These results indicate that chronic CNO-mediated hM3Dq DREADD activation of CamKIIα-positive forebrain excitatory neurons in the juvenile window does not program any persistent changes in anxiety- and despair-like behavior.

In order to test the effects of chronic CNO-mediated hM3Dq DREADD activation of CamKIIα-positive forebrain excitatory neurons in adulthood, we treated adult CamKIIα-tTA::TetO-hM3Dq bigenic male mice (3–4 months of age) with CNO (1 mg/kg; i.p.; ACNO) or vehicle once daily for thirteen days (*Figure 3K*; *Figure 3—figure supplement 1F*), following which we performed behavioral assays. Behavioral testing was carried out at two time windows of the treatment regime. The first round of behavioral testing was conducted during and soon after the cessation of CNO treatment to assess any immediate consequences on anxiety-like behavior (*Figure 3—figure supplement 2A*). The second phase of behavioral testing commenced after a three-month washout period to assess long-lasting consequences of chronic CNO-mediated hM3Dq DREADD activation of forebrain excitatory neurons in adulthood on anxiety- and despair-like behavior (*Figure 3K*; *Figure 3—figure supplement 1F*).

The first phase of behavioral testing involved assays for anxiety-like behavior on the OFT, EPM and LD avoidance test during and soon after the cessation of CNO treatment. OFT was performed

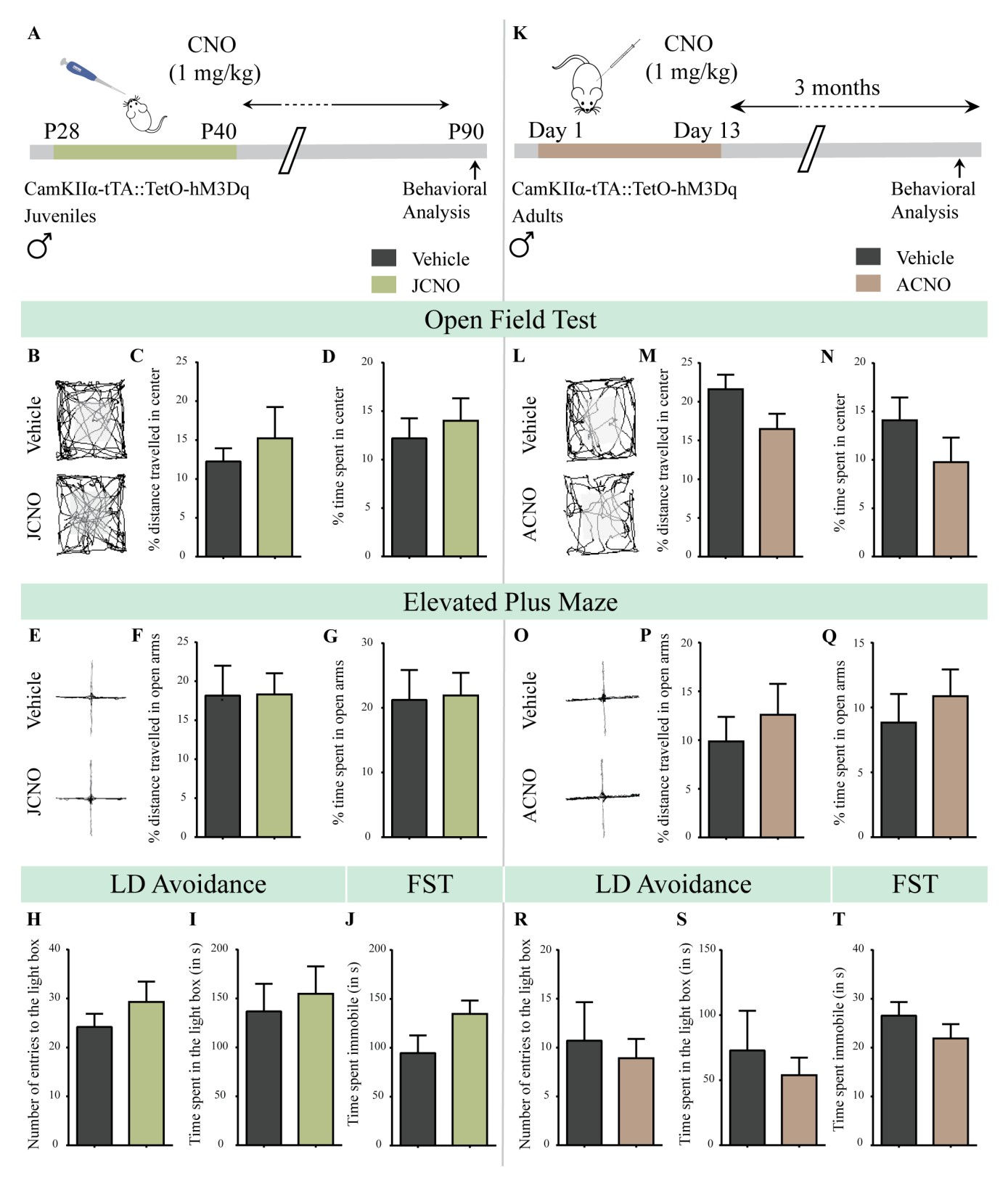

**Figure 3.** Chronic chemogenetic activation of CamKIIα-positive forebrain excitatory neurons during the juvenile window or in adulthood does not evoke any long-lasting changes in anxiety- and despair-like behavior in male mice. (**A**) Shown is a schematic of the experimental paradigm to induce chronic CNO-mediated hM3Dq DREADD activation in CamKIIα-positive forebrain excitatory neurons using bigenic CamKIIα-tTA::TetO-hM3Dq juvenile

*Figure 3 continued on next page*

*Figure 3 continued*

male mice that were fed CNO (JCNO; 1 mg/kg) or vehicle from P28-P40 and then left undisturbed till 3 months of age prior to behavioral analysis. (B) Shown are representative tracks of vehicle or JCNO-treated CamKIIα-tTA::TetO-hM3Dq bigenic adult male mice in the open field test (OFT). A history of chronic juvenile hM3Dq DREADD activation of CamKIIα-positive forebrain excitatory neurons does not evoke any persistent change in anxiety-like behavior on the OFT in adulthood, with no difference observed in the percent distance traveled in the center (C) and the percent time spent in the center (D) of the OFT arena in JCNO-treated mice as compared to vehicle-treated controls (n = 11 for vehicle; n = 8 for JCNO). (E) Shown are representative tracks of vehicle or JCNO-treated CamKIIα-tTA::TetO-hM3Dq bigenic adult mice on the elevated plus maze (EPM). Adult mice with chronic juvenile hM3Dq DREADD activation of CamKIIα-positive forebrain excitatory neurons did not exhibit any persistent changes in anxiety-like behavior on the EPM, with no change observed for the percent distance traveled (F) and the percent time spent (G) in the open arms of the EPM arena across treatment groups (n = 11 for vehicle; n = 10 for JCNO). Anxiety-like behavior was also assessed on the light-dark avoidance test and no significant alterations were seen in the number of entries to the light box (H) or the time spent in the light box (I) across treatment groups (n = 11 for vehicle; n = 10 for JCNO). A history of chronic juvenile hM3Dq DREADD activation of CamKIIα-positive forebrain excitatory neurons did not evoke any persistent change in despair-like behavior in the forced swim test (FST), with no difference observed in the time spent immobile (J) in JCNO-treated mice as compared to vehicle-treated controls (n = 11 per group). (K) Shown is a schematic of the experimental paradigm to induce chronic CNO-mediated hM3Dq DREADD activation in CamKIIα-positive forebrain excitatory neurons using bigenic CamKIIα-tTA::TetO-hM3Dq adult male mice (3–4 months of age) that received CNO (ACNO; 1 mg/kg) or vehicle via intraperitoneal administration (once daily for thirteen days) and were left undisturbed for 3 months prior to behavioral analysis. (L) Shown are representative tracks of vehicle or ACNO-treated CamKIIα-tTA::TetO-hM3Dq bigenic adult male mice in the open field test (OFT). A history of chronic hM3Dq DREADD activation of CamKIIα-positive forebrain excitatory neurons in adulthood does not evoke any persistent change in anxiety-like behavior on the OFT, with no difference observed in the percent distance traveled in the center (M) and the percent time spent in the center (N) of the OFT arena in ACNO-treated mice as compared to vehicle-treated controls (n = 7 for vehicle; n = 11 for ACNO). (O) Shown are representative tracks of vehicle or ACNO-treated CamKIIα-tTA::TetO-hM3Dq bigenic adult mice on the elevated plus maze (EPM). Chronic hM3Dq DREADD activation of CamKIIα-positive forebrain excitatory neurons in adulthood did not evoke any persistent changes in anxiety-like behavior on the EPM, with no change observed for the percent distance traveled (P) and the percent time spent (Q) in the open arms of the EPM arena across treatment groups (n = 7 for vehicle; n = 11 for ACNO). Anxiety-like behavior was also studied on the light-dark avoidance test and no change was observed in the number of entries to the light box (R) or the time spent in the light box (S) across treatment groups (n = 7 for vehicle; n = 11 for ACNO). Chronic hM3Dq DREADD activation of CamKIIα-positive forebrain excitatory neurons in adulthood did not evoke any persistent change in despair-like behavior in the forced swim test (FST), with no difference observed in the time spent immobile (T) in ACNO-treated mice as compared to vehicle-treated controls (n = 10 for vehicle; n = 9 for ACNO). Results are expressed as the mean ± S.E.M.

The online version of this article includes the following source data and figure supplement(s) for figure 3:

Source data 1. Source data for *Figure 3*.
Figure supplement 1. Chronic chemogenetic activation of CamKIIα-positive forebrain excitatory neurons during the juvenile window or in adulthood does not evoke any long-lasting changes in anxiety-like behavior in male mice.
Figure supplement 2. Chronic chemogenetic activation of CamKIIα-positive forebrain excitatory neurons in adulthood does not evoke any long-lasting changes in anxiety-like behavior in male mice during or soon after cessation of CNO treatment.

on Day 8 while the chronic CNO treatment was ongoing, and the EPM and LD avoidance test were carried out on Day 15 and Day 22, respectively, soon after cessation of CNO treatment (*Figure 3— figure supplement 2A*). No change in anxiety-like behavior was observed on the OFT (*Figure 3— figure supplement 2B–F*), EPM (*Figure 3—figure supplement 2G–K*), and LD avoidance test (*Figure 3—figure supplement 2L–N*) in the ACNO treatment group, during and soon after the cessation of CNO treatment. In the adult CNO treatment regime we did not subject mice to behavioral testing for despair-like behavior on the FST immediately after treatment, as swimming can serve as a strong stressor (*Can, 2011*; *Yankelevitch-Yahav et al., 2015*), and we intended to assess for anxiety- and despair-like behavior following a three-month washout period in the same cohort. These findings indicate that chronic CNO-mediated hM3Dq DREADD activation of CamKIIα-positive forebrain excitatory neurons in adulthood does not evoke any change in anxiety-like behavior, during or in the short duration after the cessation of CNO treatment.

The second phase of behavioral testing involved assessing for the long-lasting consequences of chronic CNO-mediated hM3Dq DREADD activation of CamKIIα-positive forebrain excitatory neurons in adulthood, with behavioral tests for anxiety- and despair-like behavior commencing 3 months post cessation of CNO treatment (*Figure 3K*; *Figure 3—figure supplement 1F*). We did not observe any change in anxiety-like behavior on the OFT in the ACNO-treated bigenic adult male mice (*Figure 3L*), with no change in the percent distance traveled in center (*Figure 3M*), percent time spent in the center (*Figure 3N*), number of entries to the center (*Figure 3—figure supplement 1G*), and the total distance traveled in the OFT arena (*Figure 3—figure supplement 1H*). Behavioral analysis of the EPM (*Figure 3O*), indicated that the ACNO-treated group did not differ in the percent distance traveled in the open arms (*Figure 3P*), percent time spent in the open arms

(*Figure 3Q*), number of entries to the open arms (*Figure 3—figure supplement 1I*), and the total distance traveled in the EPM (*Figure 3—figure supplement 1J*). Similarly, we did not observe any change in anxiety-like behavior in the LD avoidance test with no difference noted in the number of entries to the light box (*Figure 3R*) and the time spent in the light box (*Figure 3S*) across treatment groups. Further, we subjected ACNO and vehicle-treated bigenic adult male mice to the FST to assess for despair-like behavior, and noted no difference in the immobility time (*Figure 3T*). These observations reveal that chronic CNO-mediated hM3Dq DREADD activation of CamKIIα-positive forebrain excitatory neurons in adulthood does not result in any long-lasting consequences in anxiety- and despair-like behavior.

These observations collectively underscore the critical importance of the postnatal window in the long-term programming of anxiety- and despair-like behavior, as chronic CNO-mediated hM3Dq DREADD activation of CamKIIα-positive forebrain excitatory neurons is sufficient to establish persistent changes in these behaviors only when administered during the postnatal window, with no such effect noted when the same chemogenetic activation is performed either in juvenile life or in adulthood.

## Chronic chemogenetic activation of CamKIIα-positive forebrain excitatory neurons during the early postnatal window programs impaired sensorimotor gating in adulthood

Given prior evidence that a dysregulation of cortical excitation/inhibition balance during postnatal life contributes to the establishment of endophenotypes linked to schizophrenia (*Rosen et al., 2015*), we next sought to examine whether chronic CNO-mediated hM3Dq DREADD activation of CamKIIα-positive forebrain excitatory neurons in the early postnatal window influenced sensorimotor gating behavior in adulthood (*Figure 4A*). In order to assess for sensorimotor gating, we subjected CamKIIα-tTA::TetO-hM3Dq bigenic adult male mice with a history of PNCNO treatment to the prepulse inhibition (PPI) test (*Figure 4B*). We did not observe any significant alterations in the basal startle response across treatment groups (*Figure 4C*). Strikingly, we noticed a significant PPI deficit at all prepulse tones, with a decline in percent PPI to tone (120 dB) following a prepulse of + 4 dB (69 dB; *Figure 4D*, $F_{1, 19}$ = 2.063, p=0.024), + 8 dB (73 dB; *Figure 4D*, $F_{1, 21}$ = 1.136, p=0.017), and + 16 dB (81 dB; *Figure 4D*, $F_{1, 22}$ = 2.924, p=0.041, n = 12/ group) above the background noise in PNCNO-treated bigenic adult male mice. These findings indicate that chronic CNO-mediated hM3Dq DREADD activation of CamKIIα-positive forebrain excitatory neurons during the early postnatal window results in long-lasting deficits in sensorimotor gating.

Next, we attempted to understand whether chronic CNO-mediated hM3Dq DREADD activation of CamKIIα-positive forebrain excitatory neurons in the juvenile time window or in adulthood can exert similar long-term effects on sensorimotor gating behavior. CamKIIα-tTA::TetO-hM3Dq bigenic male mice (Juvenile group: P28-40; Adult group: 3–4 months of age) were administered CNO (1 mg/kg) or vehicle treatment once daily for thirteen days (*Figure 4—figure supplements 1A* and *2A*). Behavioral testing for sensorimotor gating on the PPI test (*Figure 4—figure supplements 1B* and *2B*) commenced post a three-month washout period for both the JCNO and ACNO experiments. We did not observe any significant change in the basal startle response (*Figure 4—figure supplements 1C* and *2C*) in either the JCNO or ACNO treatment groups as compared to their respective vehicle-treated controls. Further, we noted no significant difference in percent PPI in the JCNO or ACNO bigenic adult male mice to a 120 dB tone at all prepulse tones above the background noise (*Figure 4—figure supplements 1D* and *2D*).

These behavioral experiments reveal that chronic CNO-mediated hM3Dq DREADD activation of CamKIIα-positive forebrain excitatory neurons programs long-lasting changes in sensorimotor gating when the hM3Dq DREADD activation is performed in the postnatal window, but not in either the juvenile time window or in adulthood.

Chronic chemogenetic activation of CamKIIα-positive forebrain excitatory neurons during the early postnatal window results in long-lasting alterations in neuronal metabolic rate in the hippocampus and cortex.

Dysregulation of glutamatergic and GABAergic neurotransmission within forebrain neurocircuitry, including the hippocampus and several cortical regions, is thought to causally contribute to the pathogenesis of several mood-related disorders including anxiety, major depression, and schizophrenia (*Sanacora et al., 2012*; *Kendell et al., 2005*; *Choudary et al., 2005*; *Duman et al., 2019*). In

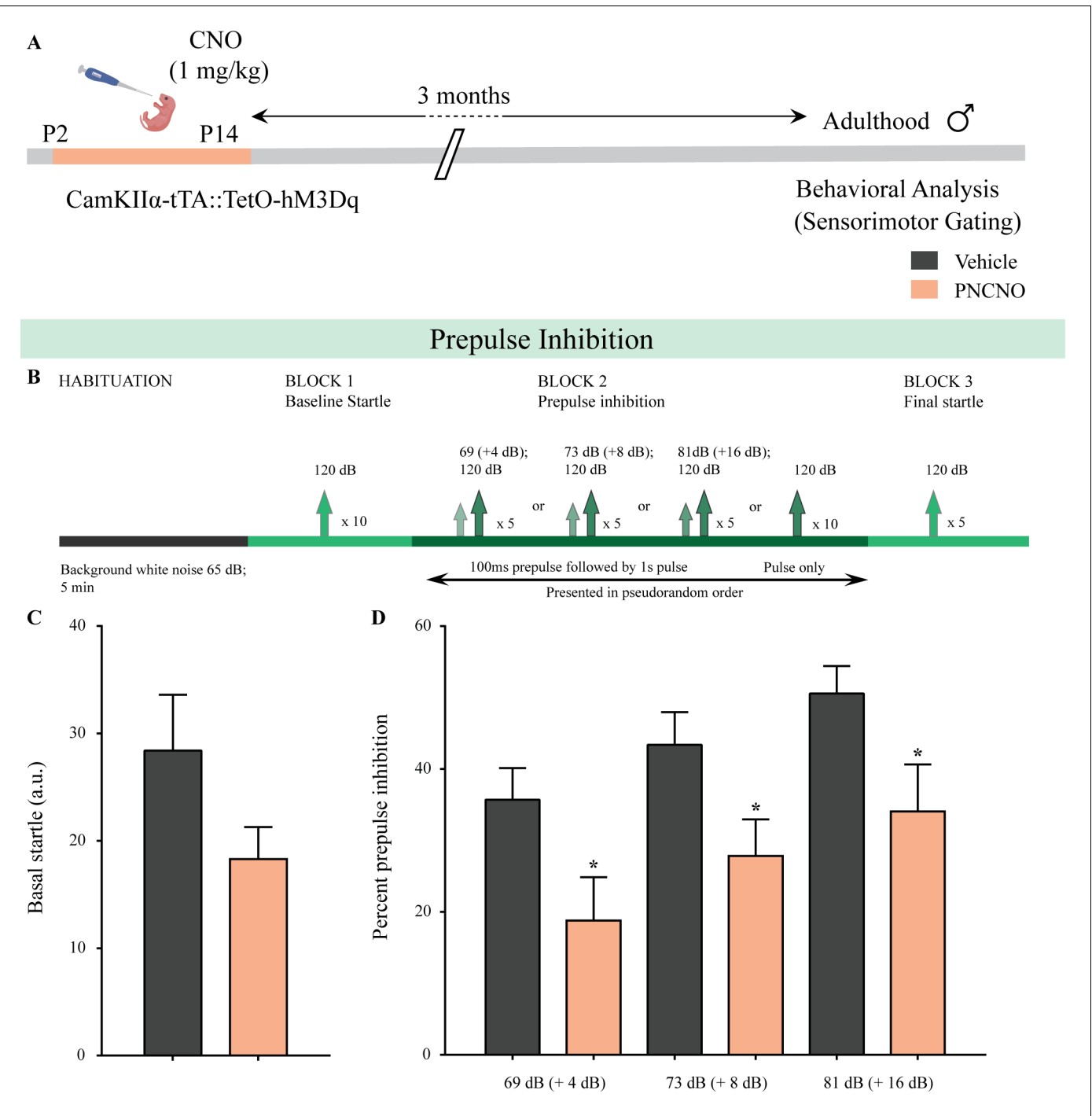

**Figure 4.** Chronic chemogenetic activation of CamKIIα-positive forebrain excitatory neurons during the early postnatal window evokes impaired sensorimotor gating in adulthood. (A) Shown is a schematic of the experimental paradigm to induce chronic CNO-mediated hM3Dq DREADD activation in CamKIIα-positive forebrain excitatory neurons using bigenic CamKIIα-tTA::TetO-hM3Dq mouse pups that were fed CNO (PNCNO; 1 mg/kg) or vehicle from P2 to P14 and then left undisturbed for 3 months prior to behavioral analysis for sensorimotor gating, using the prepulse inhibition (PPI) test, in adulthood on male mice. (B) Shown is the test paradigm for PPI to assess sensorimotor gating. Adult male mice with a history of postnatal CNO administration (PNCNO) did not show any significant change in basal startle response (C) as compared to vehicle-treated controls. PNCNO-treated adult mice exhibited deficits in sensorimotor gating, as noted via a significant decrease in prepulse inhibition (PPI)(D) at + 4 dB (69 dB), + 8 dB (73 dB), and + 16 dB (81 dB) above background noise (65 dB) when compared to vehicle-treated controls (n = 12 per group). Results are expressed as the mean ± S.E.M. *p<0.05, as compared to vehicle-treated controls using two-tailed, unpaired Student's *t*-test.

The online version of this article includes the following figure supplement(s) for figure 4:

*Figure 4 continued on next page*

*Figure 4 continued*

**Figure supplement 1.** Chronic chemogenetic activation of CamKIIα-positive forebrain excitatory neurons during the juvenile window does not alter sensorimotor gating behavior.

**Figure supplement 2.** Chronic chemogenetic activation of CamKIIα-positive forebrain excitatory neurons in adulthood does not alter sensorimotor gating behavior.

particular, metabolic dysfunction of glutamate and GABA systems are considered to be important endophenotypes of mood-related disorders (*Veeraiah et al., 2014*; *Sekar et al., 2019*; *Godfrey et al., 2018*; *Hasler and Northoff, 2011*). Hence, we next sought to investigate the effects of chronic CNO-mediated hM3Dq DREADD activation of CamKIIα-positive forebrain excitatory neurons during the early postnatal window on the metabolic activity in glutamatergic and GABAergic neurons in the hippocampus and cortex in adulthood. We orally administered CNO (PNCNO; 1 mg/kg) or vehicle to CamKIIα-tTA::TetO-hM3Dq bigenic mouse pups once daily from P2 to P14, and performed metabolic analysis in adulthood using a trace approach by infusing [1,6-$^{13}C_2$]glucose (*Figure 5A*; *Figure 5—figure supplements 1*, *2A* and *3A*). [1,6-$^{13}C_2$]Glucose is transported and metabolized in the brain to Pyruvate$_{C3}$ via glycolysis. The pyruvate$_{C3}$ is subsequently oxidized through the TCA cycles of glutamatergic and GABAergic neurons, and astrocytes to produce $^{13}$C labeled metabolites (*Figure 5—figure supplement 1*). The $^{13}$C labeling of brain metabolites was measured in $^{1}$H-[$^{13}$C]-NMR spectra of brain tissue extracts. The metabolic rate of glucose oxidation in excitatory and inhibitory neurons was determined by using the three-compartment metabolic rate model (*Patel et al., 2005*; *Tiwari et al., 2013*; *Saba et al., 2017*).

First, we measured the concentration of different metabolites in the hippocampus and cortex from the non-edited $^{1}$H-[$^{13}$C]-NMR spectrum using [2-$^{13}$C]glycine as the reference (*Figure 5—figure supplement 2B*). We did not observe any significant difference in the levels of glutamate, GABA, glutamine, aspartate, N-acetylaspartate, lactate, inositol, taurine, choline and creatine in the hippocampus and cerebral cortex of PNCNO-treated bigenic adult male mice as compared to their vehicle-treated controls (*Table 2*). We observed a significant decline in the levels of alanine in the hippocampus ($F_{1, 12}$ = 2.012, p=0.047), but not in the cortex, of bigenic adult male mice with a history of PNCNO treatment (*Table 2*).

Further, we measured the $^{13}$C labeling of amino acids from [1,6– (*Ogle et al., 2015*) $_{C2}$]glucose TCA from the $^{13}$C edited spectrum ($^{13}$C only) (*Figure 5—figure supplement 2C*). The metabolic rate glucose oxidation in excitatory and inhibitory neurons from the hippocampus and cortex was determined from the $^{13}$C label trapped into different amino acids (*Patel et al., 2005*; *Mishra et al., 2018*). Bigenic adult male mice with a history of PNCNO treatment exhibited an elevated rate of hippocampal glutamate and GABA synthesis from [1,6 - $^{13}C_2$]glucose as revealed by a significant increase in the concentration of $^{13}$C labeled Glu$_{C4}$ (*Figure 5B*, $F_{1, 12}$ = 1.335, p=0.05), GABA$_{C2}$ (*Figure 5E*, $F_{1, 12}$ = 2.4, p=0.045) and the metabolic rate of glucose oxidation in GABAergic neurons (*Figure 5G*, $F_{1, 12}$ = 1.105, p=0.001) in the hippocampus of PNCNO-treated adult mice. We did not note any difference in the concentration of $^{13}$C labeled Glu$_{C3}$ (*Figure 5C*), GABA$_{C4}$ (*Figure 5F*), and the metabolic rate of glucose oxidation in glutamatergic neurons (*Figure 5D*) in the hippocampus. We also observed an overall increase in total neuronal metabolic rate of glucose oxidation in the hippocampus of PNCNO-treated mice as compared to vehicle-treated controls (*Figure 5H*, $F_{1, 12}$ = 1.277, p=0.05). PNCNO-treated mice also showed a trend toward an increase in the concentration of $^{13}$C labeled Asp$_{C3}$ (*Figure 5—figure supplement 3C*, $F_{1, 12}$ = 1.233, p=0.07) with no change noted in the concentration of $^{13}$C labeled Gln$_{C4}$ (*Figure 5—figure supplement 3B*).

In the cortex, PNCNO-treated adult mice significantly higher levels of $^{13}$C-labeled metabolites Glu$_{C4}$ (*Figure 5I*, $F_{1, 12}$ = 2.609, p=0.009), Glu$_{C3}$ (*Figure 5J*, $F_{1, 12}$ = 1.833, p=0.026) from [1,6-$^{13}C_2$] glucose, and the metabolic rate of glucose oxidation in glutamatergic neurons of the cortex (*Figure 5K*, $F_{1, 12}$ = 2.167, p=0.007). We observed a trend toward an increase in levels of $^{13}$C-labeled metabolite GABA$_{C2}$ (*Figure 5L*, $F_{1, 12}$ = 2.521, p=0.09) and the metabolic rate of glucose oxidation in GABAergic neurons of the cortex (*Figure 5N*, $F_{1, 12}$ = 1.523, p=0.067), with no change noted in $^{13}$C-labeled GABA$_{C4}$ (*Figure 5M*) in the PNCNO-treatment group. There was a significant increase in the overall neuronal metabolic rate of glucose oxidation in the cortex (*Figure 5O*, $F_{1, 12}$ = 1.884, p=0.012) of the PNCNO-treatment group.

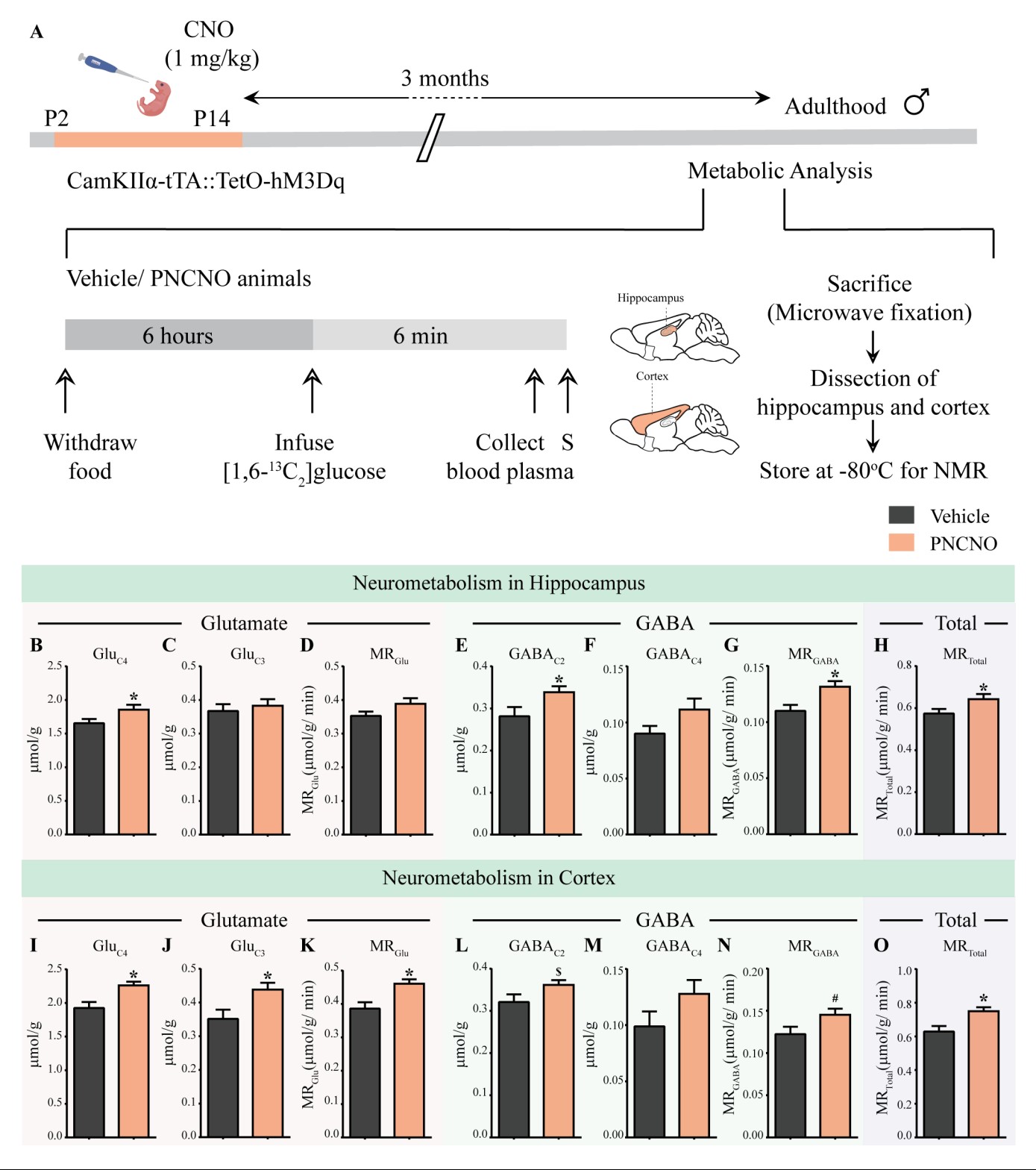

**Figure 5.** Chronic chemogenetic activation of CamKIIα-positive forebrain excitatory neurons during the early postnatal window results in long-lasting alterations in neurotransmitter cycling flux and neuronal metabolic rate in hippocampus and cortex. (**A**) Shown is a schematic of the experimental paradigm to induce chronic CNO-mediated hM3Dq DREADD activation in CamKIIα-positive forebrain excitatory neurons using bigenic CamKIIα-tTA:: TetO-hM3Dq mouse pups that were fed CNO (PNCNO; 1 mg/kg) or vehicle from P2 to P14 and then left undisturbed for 3 months prior to metabolic analysis performed in adulthood on male mice using $^1$H-[$^{13}$C]-NMR spectroscopy. Adult male mice, with a history of PNCNO or vehicle administration,

*Figure 5 continued on next page*

*Figure 5 continued*

were subjected to fasting for 6 hr, following which [1,6-$^{13}C_2$]glucose was infused via the tail-vein. Blood plasma was collected and mice were sacrificed 6 min following glucose infusion, followed by dissection of hippocampus and cortex for NMR analysis. PNCNO-treated mice exhibited a higher rate of glutamate and GABA synthesis from [1,6-$^{13}C_2$]glucose in the hippocampus as revealed by significantly higher levels of $^{13}$C-labeled metabolites Glu$_{C4}$ (B) and GABA$_{C2}$ (E) as compared to vehicle-treated controls. Levels of $^{13}$C-labeled metabolites Glu$_{C3}$ (C) and GABA$_{C4}$ (F) was not altered across treatment groups in the hippocampus. No significant change was observed in the metabolic rate of glucose oxidation in glutamatergic neurons of the hippocampus (D) across treatment groups. There was a significant increase in the metabolic rate of glucose oxidation in GABAergic neurons of the hippocampus (G) and the overall neuronal metabolic rate of the hippocampus (H), in PNCNO-treated mice as compared to vehicle-treated controls. PNCNO-treated mice had a higher rate of glutamate and GABA synthesis from [1,6-$^{13}C_2$]glucose in the cortex as revealed by significantly higher levels of $^{13}$C-labeled metabolites Glu$_{C4}$ (I), Glu$_{C3}$ (J), and the metabolic rate of glucose oxidation in glutamatergic neurons of the cortex (K) as compared to vehicle-treated controls. There was a trend toward an increase in levels of $^{13}$C-labeled metabolite GABA$_{C2}$ (L) and the metabolic rate of glucose oxidation in GABAergic neurons of the cortex (N) in the PNCNO-treated mice as compared to the vehicle-treated controls. The levels of $^{13}$C-labeled GABA$_{C4}$ (M) was not altered across treatment groups. There was a significant increase in the overall neuronal metabolic rate of the cortex (O), in PNCNO-treated mice as compared to vehicle-treated controls (n = 7 per group). Results are expressed as the mean ± S.E.M. *p<0.05, $^\$$p=0.08, $^\#$p=0.06; as compared to vehicle-treated controls using the two-tailed, unpaired Student's *t*-test.

The online version of this article includes the following figure supplement(s) for figure 5:

**Figure supplement 1.** Schematic of $^{13}$C labeling of various metabolites from [1,6-$^{13}C_2$]glucose in a three-compartment metabolic model [1,6-$^{13}C_2$] glucose is converted to Pyruvate$_{C3}$ via glycolysis and subsequently enters the TCA cycle.

**Figure supplement 2.** Representative $^1$H-[$^{13}$C]-NMR spectra from the cortex of vehicle and PNCNO-treated CamKIIα-tTA::TetO-hM3Dq adult male mice.

**Figure supplement 3.** Influence of chronic chemogenetic activation of CamKIIα-positive forebrain excitatory neurons during the early postnatal window on the levels of other $^{13}$C-labeled metabolites in the hippocampus and cortex in adult male mice.

Taken together, our data suggest a long-lasting increase in metabolic rate of neuronal glucose oxidation within the hippocampus and cortex following chronic CNO-mediated hM3Dq DREADD activation of CamKIIα-positive forebrain excitatory neurons during the early postnatal window. This suggests a persistent alteration in glutamatergic and GABAergic neurotransmission in the forebrain in adulthood as a consequence of postnatal chronic CNO-mediated hM3Dq DREADD activation of CamKIIα-positive forebrain excitatory neurons. To examine the influence of hM3Dq DREADD activation of forebrain excitatory neurons on neuronal activity, we focused on the hippocampus for the subsequent experiments.

Chronic chemogenetic activation of CamKIIα-positive forebrain excitatory neurons during the early postnatal window results in a long-lasting reduction in neuronal activity-related gene expression, and in c-Fos immunopositive cell numbers, in the adult hippocampus.

**Table 2.** Influence of chronic chemogenetic activation of CamKIIα-positive forebrain excitatory neurons during the early postnatal window on the total levels of metabolites relative to [2-$^{13}$C]glycine in the hippocampus and cortex in adult male mice.

**[1,6-$^{13}$C]glucose Infusion**

**Concentration of brain metabolites determined relative to [2-$^{13}$C]glycine (µmol/g)**

| Brain Region | Group | Glu | GABA | Gln | Asp | NAA | Ala | Lac | Ino | Tau | Cho | Cre |
|---|---|---|---|---|---|---|---|---|---|---|---|---|
| Hippocampus | Vehicle | 13.5 ± 0.3 | 3.6 ± 0.2 | 5.0 ± 0.1 | 2.3 ± 0.1 | 8.9 ± 0.1 | 0.7 ± 0.1 | 3.0 ± 0.6 | 7.3 ± 0.2 | 8.6 ± 0.2 | 2.3 ± 0.1 | 13.5 ± 0.2 |
| | PNCNO | 13.7 ± 0.3 | 3.7 ± 0.2 | 4.8 ± 0.1 | 2.5 ± 0.1 | 9.0 ± 0.1 | 0.6 ± 0.1* | 2.3 ± 0.2 | 7.5 ± 0.2 | 8.5 ± 0.2 | 2.3 ± 0.1 | 13.5 ± 0.3 |
| Cortex | Vehicle | 13.5 ± 0.3 | 3.4 ± 0.1 | 4.7 ± 0.1 | 2.9 ± 0.1 | 9.5 ± 0.4 | 0.7 ± 0.1 | 7.0 ± 1.3 | 6.0 ± 0.2 | 9.1 ± 0.1 | 2.0 ± 0.1 | 12.9 ± 0.1 |
| | PNCNO | 14.0 ± 0.2 | 3.3 ± 0.1 | 4.5 ± 0.1 | 3.2 ± 0.1$^\$$ | 9.6 ± 0.4 | 0.6 ± 0.1 | 5.3 ± 0.5 | 6.0 ± 0.1 | 9.1 ± 0.2 | 2.1 ± 0.1 | 12.9 ± 0.2 |

Chronic CNO-mediated hM3Dq DREADD activation in CamKIIα-positive forebrain excitatory neurons was achieved using bigenic CamKIIα-tTA::TetO-hM3Dq mouse pups that were fed CNO (PNCNO; 1 mg/kg) or vehicle from P2 to P14 and then left undisturbed for 3 months prior to metabolic analysis performed in adulthood on male mice and NMR spectroscopy on the hippocampus and cortex was performed to acquire $^1$H-[$^{12}$C + $^{13}$C] spectra. The concentration of metabolites was determined relative to [2-$^{13}$C]glycine. Glu: Glutamate; GABA: γ-aminobutyric acid; Gln: Glutamine; Asp: Aspartate; NAA: N-acetylaspartate; Suc: Succinate; Ala: Alanine; Lac: Lactose; Ino; Inositol; Tau: Taurine; Cho: Choline; Cre: Creatine. Results are expressed as the mean ± S.E.M (n = 7 per group). *p<0.05, $^\$$p=0.06; as compared to vehicle-treated controls using the two-tailed, unpaired Student's *t*-test.

In order to investigate the influence of chronic CNO-mediated hM3Dq DREADD activation of CamKIIα-positive forebrain excitatory neurons during the early postnatal window on hippocampal neuronal activity, we adopted two complementary approaches. First, we performed qPCR analysis for neuronal activity and plasticity-related gene expression in hippocampi derived from PNCNO and vehicle-treated CamKIIα-tTA::TetO-hM3Dq bigenic adult male mice (*Figure 6A*; *Gatto and Broadie, 2010*; *Loebrich and Nedivi, 2009*). We observed a significant decline in the expression of several neuronal activity-regulated genes namely *Fos*, *Erk1*, *Npas4*, *Staufen1*, *Staufen2*, *Nrxn1*, *Gphn*, *Shank*, *Psd95*, *Fmrp*, and *Synapsin1b* in the hippocampi derived from bigenic adult male mice with a history of PNCNO treatment (*Figure 6B*). We did not observe any alteration in expression levels of *Nr4a1*, *Junb*, *Nlgn1*, *Nlgn2*, *Mecp2*, and *Mef2c* across treatment groups (*Figure 6B*). The second approach we took was to perform cell counting analysis of c-Fos immunoreactive cell numbers within the hippocampal subfields namely, CA1, CA3, dentate gyrus (DG), and the hilus of PNCNO and vehicle administered bigenic adult male mice (*Figure 6A*). We observed a significant decline in c-Fos immunopositive cell number within the CA1 (*Figure 6C*, $F_{1, 20}$ = 3.154, p=0.004) and CA3 (*Figure 6D*, $F_{1, 20}$ = 1.67, p=0.012) subfields of the hippocampus in the PNCNO-treatment group. We did not note any change in c-Fos immunopositive cell numbers in the DG subfield (*Figure 6E*), and in the hilus (*Figure 6F*) in the PNCNO group. Collectively, our findings provide evidence that chronic CNO-mediated hM3Dq DREADD activation of CamKIIα-positive forebrain excitatory neurons during the early postnatal window can program a persistent decline in the expression of several neuronal activity and plasticity-associated genes within the hippocampus, also accompanied by a reduction in the number of c-Fos immunopositive cells suggestive of an alteration in hippocampal neuronal activity.

Chronic chemogenetic activation of CamKIIα-positive forebrain excitatory neurons during the early postnatal window alters excitatory and inhibitory spontaneous currents in the hippocampi of adult male mice.

Given that our gene expression profiling and c-Fos counting analyses pointed toward a possible change in hippocampal neuronal activity in adulthood as a consequence of postnatal hM3Dq DREADD activation of CamKIIα-positive forebrain excitatory neurons, we next performed electrophysiological studies to assess effects on hippocampal neurotransmission. Whole-cell patch clamp analysis was carried out in the somata of CA1 pyramidal neurons in acute hippocampal slices derived from PNCNO or vehicle-treated CamKIIα-tTA::TetO-hM3Dq bigenic adult male mice (*Figure 7A*; *Figure 7—figure supplements 1A* and *2A*). In order to determine the long-lasting influence of chronic hM3Dq DREADD activation of forebrain excitatory neurons during the postnatal window on intrinsic excitability in adulthood, we plotted an input-output curve by injecting increasing step currents and measured the number of action potentials (*Figure 7—figure supplement 1B*). No change was noted in the input-output curves obtained from CA1 pyramidal neurons in acute hippocampal slices derived from bigenic adult male mice with as history of PNCNO treatment (*Figure 7—figure supplement 1C*). We then measured key intrinsic membrane properties using a hyperpolarizing current step of −100 pA for 500 ms. We did not observe any change in the resting membrane potential (RMP), input resistance (RN), membrane time constant (τ), sag voltage, and accommodation index in CA1 pyramidal neurons of the PNCNO-treatment group (*Table 3*).

Measurement of sPSCs in CA1 pyramidal neurons in acute hippocampal slices derived from PNCNO-treated bigenic adult male mice revealed a significant increase in the cumulative probability of sPSC amplitude characterized by a long-tail in sPSC amplitude event distribution (*Figure 7—figure supplement 1D,E*; *Figure 7—figure supplement 2B,C*; p<0.0001), accompanied by a significant reduction in the cumulative probability of sPSC interevent intervals (*Figure 7—figure supplement 1F*; p=0.0009). Out of the total number of sPSCs analyzed, we noted events with an amplitude greater than 100 pA occurred with a significantly greater frequency in CA1 neurons from the PNCNO-treatment group (2.2%), as compared to controls (0.32%). Further, we also observed a small fraction of events (0.66%) with amplitudes greater than 250 pA in CA1 pyramidal neurons from the PNCNO-treated cohort, that were not detected in vehicle-treated controls (*Figure 7—figure supplement 2D*).

We next sought to distinguish the influence of chronic CNO-mediated hM3Dq DREADD activation of CamKIIα-positive forebrain excitatory neurons during the early postnatal window on hippocampal excitatory and inhibitory neurotransmission in adulthood. We performed whole-cell patch clamp analysis to measure sEPSCs and mEPSCs in CA1 pyramidal neurons in acute hippocampal slices derived from bigenic adult male mice with a history of PNCNO treatment (*Figure 7A*). We noted

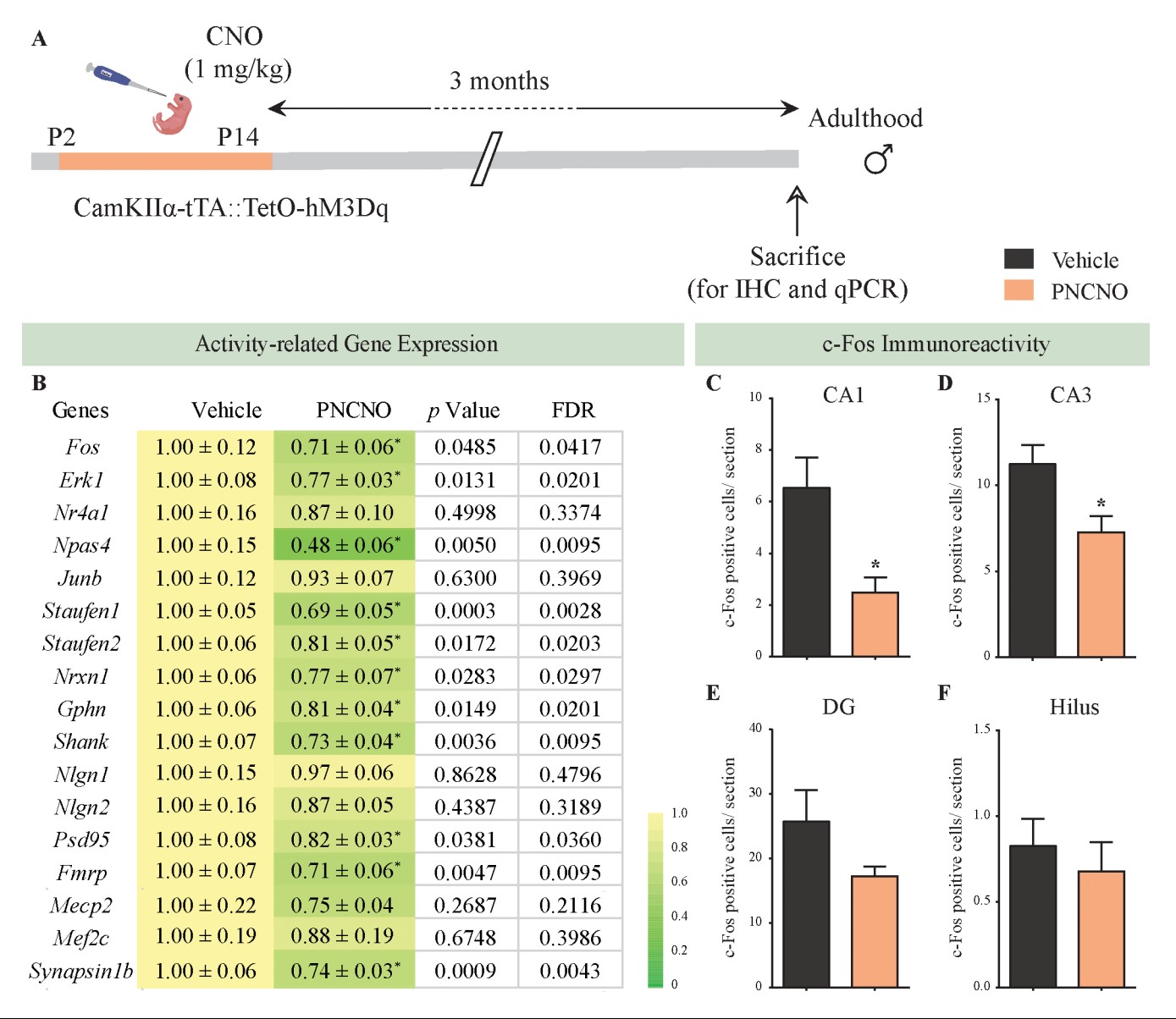

**Figure 6.** Chronic chemogenetic activation of CamKIIα-positive forebrain excitatory neurons during the early postnatal window results in a long-lasting decline in neuronal activity-related gene expression, and c-Fos immunopositive cell numbers, in the adult hippocampus. (**A**) Shown is a schematic of the experimental paradigm to induce chronic CNO-mediated hM3Dq DREADD activation in CamKIIα-positive forebrain excitatory neurons using bigenic CamKIIα-tTA::TetO-hM3Dq mouse pups that were fed CNO (PNCNO; 1 mg/kg) or vehicle from P2 to P14 and then left undisturbed for 3 months prior to qPCR analysis for neuronal activity-related gene expression and c-Fos immunohistochemistry performed in adulthood on male mice. (**B**) Shown are normalized gene expression levels for specific neuronal activity-related genes in PNCNO-treated mice represented as fold-change of their vehicle-treated controls (n = 11 per group). Heat maps indicate the extent of gene regulation. *p<0.05 as compared to vehicle-treated controls using the two-tailed, unpaired Student's *t*-test. Also shown are false-discovery rate (FDR) corrected p values. Cell counting analysis for c-Fos immunopositive cells within the hippocampus, indicated a significant reduction in the number of c-Fos positive cells/section within the CA1 (**C**) and CA3 (**D**) hippocampal subfields in PNCNO-treated mice as compared to vehicle-treated controls (n = 10 for vehicle; n = 12 for PNCNO). c-Fos immunopositive cell numbers were unaltered in the DG (**E**) and the hilar subfield (**F**) of the hippocampus in PNCNO-treated mice as compared to vehicle-treated controls. Results are expressed as the mean ± S.E.M. *p<0.05 as compared to vehicle-treated controls using the two-tailed unpaired Student's *t*-test.

The online version of this article includes the following source data for figure 6:

**Source data 1.** List of qPCR primers.

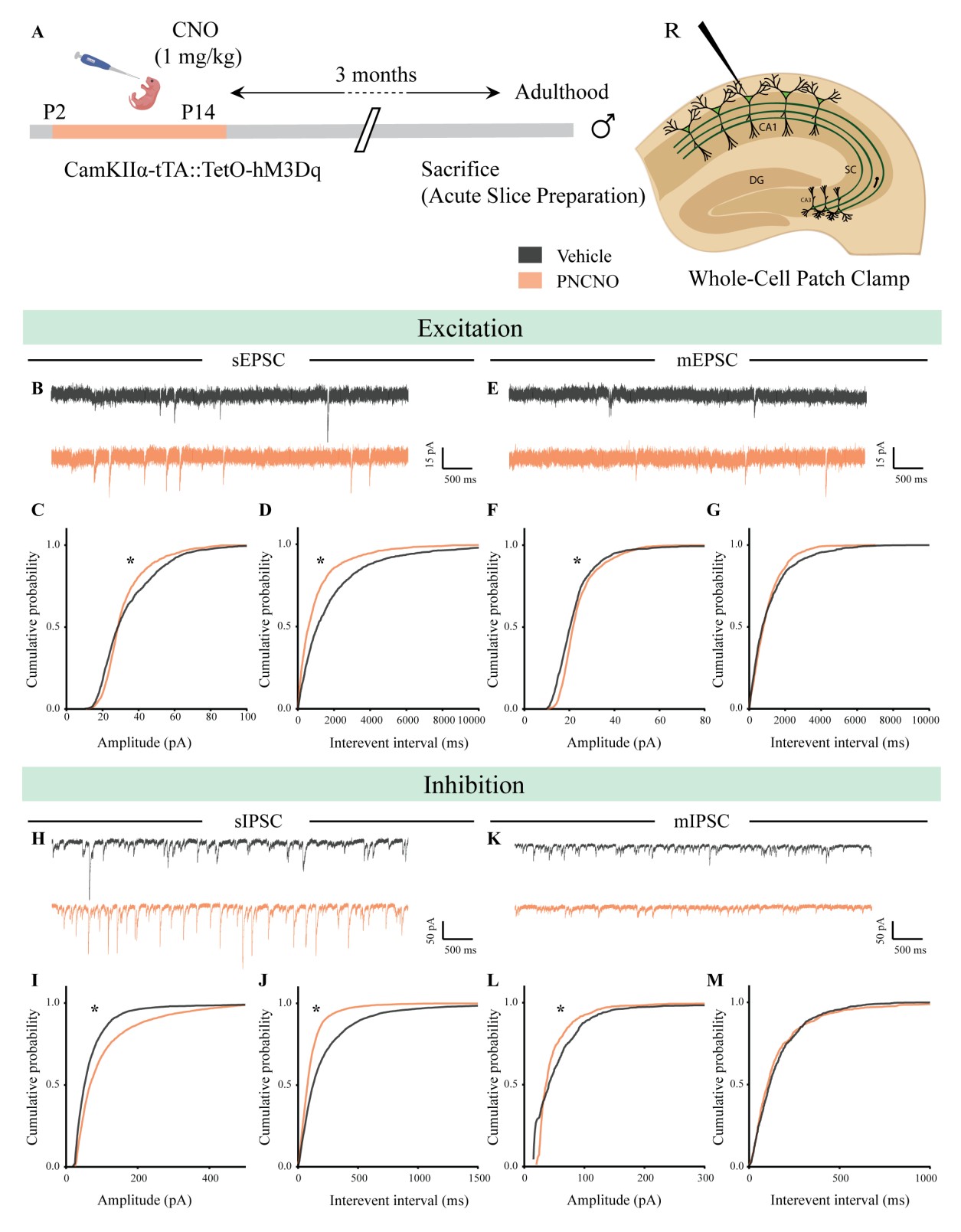

**Figure 7.** Chronic chemogenetic activation of CamKIIα-positive forebrain excitatory neurons during the early postnatal window alters excitatory and inhibitory spontaneous currents in the hippocampi of adult male mice. (**A**) Shown is a schematic of the experimental paradigm to induce chronic CNO-mediated hM3Dq DREADD activation in CamKIIα-positive forebrain excitatory neurons using bigenic CamKIIα-tTA::TetO-hM3Dq mouse pups that were fed CNO (PNCNO; 1 mg/kg) or vehicle from P2 to P14 and then left undisturbed for 3 months prior to electrophysiological analysis, in acute

*Figure 7 continued on next page*

*Figure 7 continued*

hippocampal slices derived from adult male mice. Whole-cell patch clamp was performed to record sEPSCs/mEPSCs and sIPSCs/mIPSCs in the somata of CA1 pyramidal neurons. R – Recording electrode. (B) Shown are representative sEPSC traces of CA1 pyramidal neurons from vehicle and PNCNO-treated mice. (C) PNCNO-treated mice showed significantly altered cumulative probability of sEPSC amplitude with a small increase at lower amplitudes (<30 pA) and a significant decline in large-amplitude events as compared to vehicle-treated controls. (D) PNCNO-treated mice showed a significant decline in the cumulative probability of sEPSC interevent intervals as compared to vehicle-treated controls (n = 8 cells for vehicle; n = 5 cells for PNCNO). (E) Shown are representative mEPSC traces of CA1 pyramidal neurons from vehicle and PNCNO-treated mice. (F) PNCNO-treated mice showed significantly enhanced cumulative probability of sEPSC amplitude as compared to vehicle-treated controls. (G) No significant change was observed in the cumulative probability of sEPSC interevent intervals in PNCNO-treated mice as compared to vehicle-treated controls (n = 5 cells for vehicle; n = 6 cells for PNCNO). (H) Shown are representative sIPSC traces of CA1 pyramidal neurons from vehicle and PNCNO-treated mice. PNCNO-treated mice showed a significant increase in the cumulative probability of sIPSC amplitude (I), along with a significant decline in the cumulative probability of sIPSC interevent intervals (J) as compared to vehicle-treated controls (n = 8 cells for vehicle; n = 7 cells for PNCNO). (K) Shown are representative mIPSC traces of CA1 pyramidal neurons from vehicle and PNCNO-treated mice. (L) PNCNO-treated mice showed a significant decline in the cumulative probability of mIPSC amplitude as compared to vehicle-treated controls (n = 6 cells for vehicle; n = 7 cells for PNCNO). (M) No significant change was observed in the cumulative probability of mIPSC interevent intervals across treatment groups. Results are expressed as cumulative probabilities. *p<0.001 as compared to PNCNO-treated group using the Kolmogorov-Smirnov two-sample comparison.

The online version of this article includes the following figure supplement(s) for figure 7:

**Figure supplement 1.** Chronic chemogenetic activation of CamKIIα-positive forebrain excitatory neurons during the early postnatal window does not change intrinsic excitability but alters spontaneous network activity in the hippocampi of adult male mice.

**Figure supplement 2.** Effect of chronic chemogenetic activation of CamKIIα-positive forebrain excitatory neurons during the early postnatal window on the distribution of spontaneous network events in hippocampi of adult male mice.

a significantly altered sEPSC amplitude distribution in CA1 pyramidal neurons of PNCNO-treated adult male mice (*Figure 7B,C*; p<0.0001), with a small but significant increase in low amplitude events (<30 pA), and a significant decline in large-amplitude events. Further, we observed a significant decrease in cumulative probability of sEPSC interevent intervals in CA1 pyramidal neurons from the PNCNO-treatment group (*Figure 7D*; p<0.0001). We observed a small, but significantly enhanced cumulative probability of mEPSC amplitude (*Figure 7E,F*; p<0.0001), with no change observed in the cumulative probability of mEPSC interevent intervals (*Figure 7G*) in bigenic adult male mice with a history of PNCNO treatment.

To assess effects on hippocampal inhibitory neurotransmission, we measured sIPSCs and mIPSCs in CA1 pyramidal neurons in acute hippocampal slices. We noted a significant increase in the cumulative probability of sIPSC amplitude (*Figure 7H,I*; p<0.0001), concomitant with a significant reduction in the cumulative probability of sIPSC interevent intervals (*Figure 7J*; p<0.0001) in bigenic adult male mice with a history of PNCNO treatment. Further, we noted a significant reduction in the cumulative probability of mIPSC amplitude (*Figure 7K,L*; p<0.0001), with no change noted in the cumulative probability of mIPSC interevent intervals (*Figure 7M*) in CA1 neurons of PNCNO-treated adult male mice.

Our electrophysiological studies performed on CA1 neurons in acute hippocampal slices of adult male mice with a history of chemogenetic activation of CamKIIα-positive forebrain excitatory

**Table 3.** Effect of chronic chemogenetic activation of CamKIIα-positive forebrain excitatory neurons during the early postnatal window on intrinsic membrane properties in adulthood.

| Adults | CA1 pyramidal neurons | | | | | |
|---|---|---|---|---|---|---|
| **Intrinsic properties** | | | | | | |
| Group | RMP (mV) | Input resistance (MΩ) | τ (ms) | Sag (mV) | Sag (%) | Accomodation index |
| Vehicle | −62.3 ± 0.655 | 195.1 ± 10.89 | 18.87 ± 1.574 | −4.282 ± 0.288 | 4.956 ± 0.317 | 0.375 ± 0.054 |
| PNCNO | −62.81 ± 0.682 | 180.2 ± 10.46 | 18.66 ± 1.269 | −3.603 ± 0.397 | 4.173 ± 0.420 | 0.392 ± 0.046 |

Chronic CNO-mediated hM3Dq DREADD activation of CamKIIα-positive forebrain excitatory neurons was performed using bigenic CamKIIα-tTA::TetO-hM3Dq mouse pups that were fed CNO (PNCNO; 1 mg/kg) or vehicle from P2 to P14 and then left undisturbed for 3 months prior to electrophysiological analysis, in acute hippocampal slices derived from adult male mice. Whole-cell patch clamp was performed to determine intrinsic properties in the somata of CA1 pyramidal neurons. RMP: Resting membrane potential, τ: Membrane time constant. Results are expressed as the mean ± S.E.M. (n = 18 cells for vehicle; n = 25 cells for PNCNO).

neurons during the early postnatal window demonstrates the programming of persistent increases in spontaneous network activity, accompanied by significantly altered hippocampal excitatory and inhibitory neurotransmission.

## Discussion

The major finding of our study is that chronic chemogenetic hM3Dq DREADD activation of CamKIIα-positive forebrain excitatory neurons in the first two weeks of postnatal life is sufficient to program the emergence of enhanced anxiety-, despair- and schizophrenia-like behavior in adult male mice. In contrast, chronic chemogenetic activation of CamKIIα-positive forebrain excitatory neurons in either the juvenile or adult temporal window did not result in any persistent changes in mood-related behavior. Chronic chemogenetic activation of forebrain excitatory neurons in postnatal life also resulted in persistent changes in glutamate/GABA metabolism in the hippocampus and cortex, accompanied by a long-lasting decline in hippocampal activity and plasticity-associated gene expression, and altered hippocampal spontaneous excitatory and inhibitory currents. Given prior reports that several models of early adversity exhibit enhanced signaling via Gq-coupled neurotransmitter receptors in the forebrain (*Benekareddy et al., 2010*; *Benekareddy et al., 2011*; *Malkova et al., 2014*; *Proulx et al., 2014*; *Sarkar et al., 2014b*; *Moreno et al., 2011*), our findings posit that enhanced Gq-signaling-mediated activation of forebrain excitatory neurons in the critical temporal window of postnatal life may serve as a putative mechanism to program enhanced risk for adult psychopathology, a hallmark feature of models of early adversity.

Results from multiple rodent models including maternal separation (MS), maternal neglect, and postnatal fluoxetine (PNFlx) indicate that the first two weeks of postnatal life are critical to the long-lasting programming of anxiety- and despair-like behavior (*Rebello et al., 2014*; *Suri and Vaidya, 2015*; *Roque et al., 2014*; *Freund et al., 2013*). Evidence from several of these rodent models suggests enhanced functionality of Gq-coupled neurotransmitter receptors in the forebrain (*Benekareddy et al., 2010*; *Benekareddy et al., 2011*; *Malkova et al., 2014*; *Proulx et al., 2014*; *Sarkar et al., 2014b*; *Moreno et al., 2011*). Furthermore, pharmacological studies indicate that stimulation of the Gq-coupled 5-HT$_{2A}$ receptor during the early postnatal window can program persistent mood-related behavioral changes (*Sarkar et al., 2014b*), and that 5-HT$_{2A}$ receptor blockade overlapping with MS or PNFLx can prevent the emergence of adult anxiety- and despair-like behavior (*Benekareddy et al., 2011*; *Sarkar et al., 2014b*). Our study directly tests the role of enhanced Gq-signaling-mediated activation of forebrain excitatory neurons in the early postnatal window in programming mood-related behavioral changes, and demonstrates that this perturbation when performed in the critical postnatal window, but not in juvenile or adult life, is sufficient to program the emergence of anxiety-, despair- and schizophrenia-like behaviors in adulthood.

Adult male mice with a history PNCNO treatment showed both enhanced anxiety- and despair-like behavior, whereas adult female mice exhibited enhanced anxiety-, but not despair-like behavioral changes. Sexually dimorphic effects of early adversity have been previously reported, with females suggested to be resistant to some of the behavioral consequences of early adversity, in particular the programming of despair-like behavioral changes (*Leussis et al., 2012*; *Roman et al., 2004*; *Dimatelis et al., 2016*; *Lundberg et al., 2017*; *Desgent et al., 2012*; *de Melo et al., 2018*). Our results raise the possibility of sexually dimorphic behavioral consequences of early postnatal chemogenetic activation of forebrain excitatory neurons. A caveat of our study is that the neurometabolic, electrophysiological and molecular experiments were performed only in adult male mice, with a limited battery of behavioral analysis carried out in adult females, in part due to the large numbers of bigenic mice required to be maintained for these experiments. This precluded the possibility of careful analysis of sexual dimorphism in the consequences of enhanced Gq- signaling-driven within forebrain excitatory neurons in postnatal life, which will require detailed further experimentation.

Given that thus far very few studies have used chemogenetic strategies during developmental time windows (*Teissier et al., 2020*; *Wong et al., 2018*), we characterized the consequences of hM3Dq DREADD activation of forebrain excitatory neurons in the postnatal window using both electrophysiological and biochemical approaches. Our observation of hM3Dq DREADD-mediated induction of robust spiking activity in postnatal slices parallels observations made in adulthood (*Alexander et al., 2009*; *Pati et al., 2019*). Chronic chemogenetic hM3Dq DREADD activation

during postnatal life enhanced neuronal activity marker expression in the hippocampus and cortex, as well as increased network activity, enhanced spontaneous excitatory events, and reduced spontaneous inhibitory events in CA1 pyramidal neurons in PNCNO-treated mouse pups. Our studies also indicated that chronic chemogenetic hM3Dq DREADD activation of forebrain excitatory neurons during postnatal life does not impact normal physical growth, developmental milestones such as eye opening or ontogeny of reflex development. We also addressed the possibility that off-target effects of CNO (*Gomez et al., 2017*; *MacLaren et al., 2016*) may impact our interpretations by extensively addressing effects of postnatal CNO treatment to genotype-control or background strain mouse pups, and noted no change in the emergence of anxiety- or despair-like behaviors in adulthood. These controls are particularly relevant given that very few reports have used chemogenetic perturbations during these early postnatal windows (*Teissier et al., 2020*; *Wong et al., 2018*). Our results do not allow us to completely rule out potential off-target effects of PNCNO treatment on locomotion in the OFT in adulthood. In this regard, our studies using an alternate DREADD agonist C21 to chemogenetically activate forebrain excitatory neurons during the postnatal window resulted in robust increases in anxiety-like behavior in adulthood, with no effects noted on total locomotion. This highlights the importance of using multiple DREADD ligands, especially when considering potential off-target effects of specific DREADD agonists on behavioral tasks.

We next addressed whether chronic chemogenetic activation of CamKIIα-positive forebrain excitatory neurons in postnatal life recapitulates the effects of early adversity in programming changes in schizophrenia-like behavior (*Girardi et al., 2014*; *Ellenbroek et al., 1998*), and repetitive behavior (*Malkova et al., 2012*). We noted a significant impairment in sensorimotor gating indicated by PPI deficits, but no change in stereotypic behavior, in adult mice with a history of PNCNO treatment. Deficient PPI is considered to be a behavioral deficit associated with schizophrenia-like behavior in both genetic or environmental perturbation based animal models (*Ellenbroek et al., 1998*; *Khan and Powell, 2018*; *Mena et al., 2016*; *Geyer et al., 2001*; *Belforte et al., 2010*). Preclinical genetic models targeting signaling pathways downstream to Gq (PLC-$\beta$1$^{-/-}$ mice) exhibit enhanced schizophrenia-like behavior (*McOmish et al., 2008*). Further, loss of function of the Gq-coupled mGluR5 receptor in parvalbumin-positive interneurons increased both compulsive behavior and aberrant sensorimotor gating (*Barnes et al., 2015*). It is important to note that PPI deficits are also common across various other neuropsychiatric conditions, in addition to schizophrenia (*Powell et al., 2012*; *Nestler and Hyman, 2010*). Several reports indicate that early adversity during the perinatal window results in PPI impairments in adulthood (*Ellenbroek et al., 1998*; *Ko et al., 2014*; *Smith et al., 2007*; *Fabricius et al., 2008*). However, both the intensity and timing of the early stressor could program differing outcomes on PPI (*Ellenbroek et al., 1998*; *Fabricius et al., 2008*; *Ellenbroek and Cools, 2002*). For example, severe maternal deprivation evokes robust PPI deficits, whereas short duration maternal separation has no effect on PPI (*Ellenbroek et al., 1998*; *Ellenbroek and Cools, 2002*). In this regard, our findings that chemogenetic activation of forebrain excitatory neurons produces an entire spectrum of mood-related behavioral changes, namely enhanced anxiety-, despair- and schizophrenia-like behaviors is suggestive of behavioral endophenotypes noted with the more severe of early stress models (*Nestler and Hyman, 2010*; *Bolton et al., 2017*; *Walker et al., 2017*).

Our results that the timing of the chronic chemogenetic activation of forebrain excitatory neurons is central to determining consequent changes in mood-related behavior underscores the key importance of 'critical' periods for programming emotionality (*Bock et al., 2014*; *Leonardo and Hen, 2008*). We observed no change in anxiety-, despair- and schizophrenia-like behaviors in either the juvenile or adult chronic CNO paradigms. Our treatment involved administration of the DREADD agonist CNO orally to pups/juveniles, and intraperitoneally to adult mice. Although we cannot assume that the effects observed across lifespan involved equivalent pharmacodynamics of CNO, it is noteworthy that there is no effect on anxiety-like behavior following chronic administration of CNO in both juvenile and adult CamKIIα-hM3Dq bigenic mice. This suggests that the chemogenetic activation of Gq signaling in forebrain excitatory neurons needs to be performed in the postnatal window (P2–14) to program persistent mood-related behavioral changes. While our studies do not allow us to parcellate out the exact duration of this critical window, it likely encompasses the first two weeks of life. In this regard, both the MS and PNFLx models have critical periods spanning from P2 to 14 and P2–11, respectively (*Rebello et al., 2014*; *Roque et al., 2014*; *Freund et al., 2013*). These temporal windows overlap with distinct critical periods including the stress

hyporesponsive period (*Schmidt et al., 2003*; *Levine, 2001*; *Suchecki, 2018*), a neurodevelopmental window for the refinement of multiple cortical circuits (*Hensch, 2004*; *Hensch, 2005*), including the maturation of serotonergic afferents to the cortex (*Teissier et al., 2017*; *Vitalis and Verney, 2017*), and the tuning of excitation-inhibition balance across cortical microcircuits (*Sohal and Rubenstein, 2019*; *Tatti et al., 2017*; *Xue et al., 2014*). The first two weeks of postnatal life also constitutes a window in which apoptotic cell death plays a key role in the shaping of neocortical microcircuitry (*Wong and Marín, 2019*). Activity within cortical pyramidal neurons can directly shape interneuron survival (*Wong et al., 2018*), thus influencing the manner in which the optimal balance between excitatory and inhibitory neurons in cortical microcircuits is established. A recent report indicates that hM3Dq DREADD activation of the medial prefrontal cortex during postnatal life can abrogate the influence of maternal separation on oligodendrogenesis and despair-like behavior (*Teissier et al., 2020*). However, the use of a pan-neuronal human synapsin promoter to drive the hM3Dq DREADD (*Teissier et al., 2020*), and the absence of a non-maternal separation cohort makes it difficult to directly compare with our results. Our observations support the view that chemogenetic activation of forebrain excitatory neurons from P2 to 14 could impinge on several key neurodevelopmental processes, thus establishing a substrate for the emergence of perturbed mood-related behaviors in adulthood.

Associated with the long-lasting behavioral changes programmed by chronic DREADD activation of CamKIIα-positive forebrain excitatory neurons, we noted persistent dysregulation of glutamate and GABA neurotransmitter metabolism, a decline in the expression of neuronal activity- and plasticity-related markers, as well as alterations in hippocampal spontaneous excitatory and inhibitory currents. The dysregulation of both glutamate and GABA systems is amongst the key factors in the pathophysiology of several psychiatric disorders including anxiety, depression, and schizophrenia (*Sanacora et al., 2012*; *Kendell et al., 2005*; *Duman et al., 2019*; *Brambilla et al., 2003*; *Bergink et al., 2004*). Neuroimaging studies on human subjects with mood disorders and schizophrenia demonstrate altered volume and resting-state functional activity in several forebrain regions, including hippocampus, sensory and frontal cortices (*Koike et al., 2013*; *Wolf et al., 2011*; *Kühn and Gallinat, 2013*). A major endophenotype that reflects persistent alterations in neuronal activity in mood-related disorders is the levels and neurometabolic activity of glutamate and GABA, the major excitatory and inhibitory neurotransmitters, respectively (*Sanacora et al., 2012*; *Kendell et al., 2005*; *Duman et al., 2019*; *Veeraiah et al., 2014*; *Sekar et al., 2019*; *Godfrey et al., 2018*; *Hasler and Northoff, 2011*). Although $^1$H-MRS has been widely used to examine the levels of these neurotransmitters in both human patients and rodents (*Zieminska et al., 2018*; *Dyke et al., 2017*), there has been a scarcity of studies to investigate neurometabolic activity, which represent a functional readout of metabolic dynamics in these neurocircuits (*Patel et al., 2004*; *de Graaf et al., 2003*). We employed $^1$H-[$^{13}$C]-NMR spectroscopy to measure the metabolic rate of excitatory and inhibitory neurons in conjunction with infusion of [1,6-$^{13}$C$_2$]glucose (*Patel et al., 2005*; *Tiwari et al., 2013*; *Patel et al., 2004*; *de Graaf et al., 2003*). The glutamate hypothesis of mood disorders is based on observations of elevated glutamate levels, associated with changes in glutamate receptors, biosynthetic and regulatory pathways both in human patients and rodent models of anxiety/despair-like behavior and schizophrenia (*Sanacora et al., 2012*; *Duman et al., 2019*). Consistent with this hypothesis, we observed an increase in glucose oxidation in the TCA cycle of glutamatergic neurons in the hippocampus and cortex of adult mice with a history of PNCNO treatment. The rate of neuronal glucose oxidation and neurotransmitter cycle are stoichiometrically coupled during the entire range of brain activity (*Patel et al., 2005*; *Hyder et al., 2006*; *Sibson et al., 1998*). Hence, increased neuronal glucose oxidation in PNCNO-treated mice suggests enhanced excitatory and inhibitory neurotransmission. Furthermore, we observed an increase in metabolic rate of hippocampal GABAergic neurons, and a trend toward an increase in this measure in the cortex. It is important to note that though we see an increase in metabolic rate of both glutamatergic and GABAergic neurons, a history of PNCNO treatment does not influence the neurotransmitter pool of glutamate or GABA either in the hippocampus or cortex. While our observations for enhanced glutamatergic metabolic rate in PNCNO-treated animals are consistent with clinical and preclinical reports of enhanced glutamate function in mood disorders (*Sanacora et al., 2012*; *Duman et al., 2019*), our observations with GABA differ from the reports of reduced GABA levels observed in human subjects and several adult-onset stress based rodent models of mood-related behavioral changes (*Duman et al., 2019*; *Pilc and Nowak, 2005*; *Kalueff and*

*Nutt, 2007*). Thus far, neurometabolic studies on preclinical models of early-life stress, or in patients with a life history of early adversity have not been carried out, making it difficult to directly compare our observations. Collectively, our results suggest that driving enhanced Gq-signaling-based activation of forebrain excitatory neurons in the postnatal window can evoke persistent dysregulation of amino acid neurotransmitter system metabolism, which may contribute to the long-lasting behavioral changes.

We focused our gene expression and electrophysiological studies on the hippocampus, which has been strongly implicated in mood-related disorders (*Santos et al., 2018*; *Campbell and Macqueen, 2004*; *Femenía et al., 2012*). Early stress influences both hippocampal neuronal morphology and plasticity, features that contribute to the behavioral sequelae of early trauma (*Kim et al., 2006*; *Fenoglio et al., 2006*; *Maccari et al., 2014*; *McEwen et al., 2016*). We noted a decreased expression of several activity- and plasticity-related markers within the hippocampus, observed months post the cessation of PNCNO treatment, indicative of persistent molecular changes that ensue from the transient postnatal perturbation. These markers included transcription/translation factors, scaffolding proteins, cell adhesion molecules, and ion channels previously implicated in the regulation of excitation-inhibition balance (*Gatto and Broadie, 2010*; *Loebrich and Nedivi, 2009*). These results are suggestive of the programming of an altered excitation/inhibition within the hippocampus, which is supported by our electrophysiological observations. Several studies in the past have investigated neurophysiological consequences of early adversity (*Ali et al., 2011*). Previous results indicate persistent dysregulation of signaling via Gq-coupled neurotransmitter receptors (M1 and 5-HT$_{2A}$) in the neocortex of maternally separated animals (*Benekareddy et al., 2010*; *Proulx et al., 2014*). Rodent models of rearing in an impoverished environment (*Cui et al., 2006*; *Brunson et al., 2005*), poor maternal care (*Bagot et al., 2009*; *Weaver et al., 2004*; *Meaney and Szyf, 2005*), neonatal novelty exposure (*Zou et al., 2001*), and maternal separation (*Kehoe et al., 1995*; *Gruss et al., 2008*; *Salzberg et al., 2007*) are all associated with impairment of hippocampal long-term potentiation (LTP). Further, in vivo electrophysiological recordings in a model of neonatal isolation indicate a decline in hippocampal outputs (*Bartesaghi, 2004*; *Bartesaghi et al., 2006*). Our observations of reduced expression of the activity marker c-Fos in all hippocampal subfields, concomitant with an increase in cumulative probabilities of sIPSC amplitude and a reduction of sIPSC interevent intervals, is indicative of decreased activity in hippocampal networks in keeping with observations in early stress models (*Ali et al., 2011*). Our results support an overall increase in inhibitory neurotransmission within the hippocampi of adult mice with a history of PNCNO treatment. The increase in GABA flux, enhanced metabolic rate of GABAergic neurons, and the shift toward increased inhibition noted in hippocampi of PNCNO-treated mice could arise as an adaptive compensation to increased DREADD-mediated excitation during the postnatal window. The effect on hippocampal excitatory neurotransmission on the other hand appears more complex, with an increase in low amplitude and a decline in larger amplitude spontaneous events, along with an increase in the frequency of sEPSC events. This was concomitant with an increase in the cumulative probability of mEPSC amplitude, and a decline in high amplitude mIPSC events. This suggests that the overall decline in inhibition is unlikely to be cell-autonomous, and probably emerges as a consequence of an alteration in the excitatory-inhibitory recurrent network of the hippocampus. Our results do not allow us to distinguish whether the dysfunctional glutamate/GABA metabolism and neurotransmission observed in PNCNO-treated mice serve as instructive/permissive to the development of psychopathology, or simply arise as compensatory adaptations due to increased neuronal activation of forebrain excitatory neurons in the postnatal window. This motivates future experiments to address the influence of driving Gq-signaling-mediated activation of forebrain excitatory neurons in the postnatal window on excitatory and inhibitory neurotransmission in both cortical and hippocampal networks, as well as the emergence of excitation-inhibition balance within these neurocircuits. A complementary set of studies involving the perturbation of inhibitory Gi signaling within forebrain excitatory neurons in early postnatal life, both baseline and within the background of early adversity, would also provide valuable insights into the cortical microcircuitry and signaling pathways that contribute to the programming of long-lasting behavioral changes in emotionality.

In conclusion, we show that chemogenetic activation of forebrain excitatory neurons during postnatal life evokes a long-lasting increase in anxiety-, despair-, and schizophrenia-like behavior. These behavioral changes are accompanied by a dysregulation of glutamate/GABA metabolism in the cortex and hippocampus, as well as perturbed inhibitory and excitatory neurotransmission within the

hippocampus in adulthood. Our perturbation evokes several of pathophysiological features associated with preclinical and clinical studies of early adversity. These findings suggest the intriguing possibility that early adversity could program specific aspects of long-lasting behavioral, molecular, metabolic and functional changes via a modulation of Gq-signaling-mediated neuronal activation of forebrain excitatory neurons within the critical temporal window of postnatal life.

## Materials and methods

### Animals

The CamKIIα-tTA transgenic mice (*Mayford et al., 1996*) were gifted by Dr. Christopher Pittenger, Department of Psychiatry, Yale School of Medicine. The TetO-hM3Dq mice (Cat. No. 014093; Tg (TetO-CHRM3*)1Blr/J) and C57BL/6J mice were purchased from Jackson Laboratories, USA. The genotypes of CamKIIα-tTA::TetO-hM3Dq animals were confirmed by PCR-based genotyping analysis. All experiments using bigenic mice utilized mice which were homozygous for both CamKIIα-tTA and TetO-hM3Dq. Single-positive animals, positive for either CamKIIα-tTA or TetO-hM3Dq, as well as the background strain C57BL/6J were used for control experiments. The animals were bred in the Tata Institute of Fundamental Research (TIFR) animal house facility. All animals were maintained on a 12 hr light-dark cycle (7 am to 7 pm), with ad libitum access to food and water. Slice electrophysiology experiments were carried out at the Jawaharlal Nehru Centre for Advanced Scientific Research (JNCASR), Bengaluru and $^1$H-[$^{13}$C]-NMR experiments were carried out at the Centre for Cellular and Molecular Biology (CCMB), Hyderabad. Experimental procedures were carried out as per the guidelines of the Committee for the Purpose of Control and Supervision of Experiments on Animals (CPCSEA), Government of India and were approved by the TIFR, JNCASR, and CCMB animal ethics committees. Care was taken across all experiments to minimize animal suffering and restrict the number of animals used.

### Drugs

DREADD agonist, CNO(Cat. No. 4936, Tocris, UK) was used to selectively activate the excitatory DREADD, hM3Dq. CNO was dissolved in 5% aqueous sucrose solution for oral delivery in postnatal and juvenile treatment experiments, and in physiological saline for intraperitoneal delivery in adult-onset treatments. The alternative DREADD agonist compound 21 (C21; Cat. No. 5548, Tocris, UK) was dissolved in 10 μl DMSO, and then diluted in 5% aqueous sucrose solution to a 1 ml stock solution. Following this, the solution was aliquoted and stored at −80℃ before using for oral delivery in postnatal experiments. For vehicle treatment, the base solution without the drugs was used. For slice electrophysiology experiments, stock solutions of CNO, 6-Cyano-7-nitroquinoxaline-2,3-dione disodium (CNQX disodium salt; Cat. No. 1045, Tocris, UK), DL-2-Amino-5-phosphonopentanoic acid sodium salt (AP5, Cat. No. 3693, Tocris, UK), (-)-Bicuculline methochloride (Cat. No. 0131, Tocris, UK), and Tetrodotoxin citrate (TTX; Cat. No. ab120055, Abcam, UK) were prepared and aliquots were stored at −20℃. For all slice electrophysiology experiments, acute slice preparations had bath application of drugs in artificial CSF (aCSF) using a perfusion system.

### Western blotting

To assess HA-tagged hM3Dq DREADD expression in the hippocampus and cortex of CamKIIα-tTA:: TetO-hM3Dq bigenic mice on postnatal Day 7 (P7), we performed western blotting analysis for the HA antigen. To examine the influence of CNO-mediated hM3Dq DREADD activation on expression levels of neuronal activity markers (c-Fos, phospho-ERK/ERK), we fed a single dose of 1 mg/kg CNO or vehicle to CamKIIα-tTA::TetO-hM3Dq bigenic mouse pups (P7) and sacrificed them 15 min post-feeding. In order to examine the effect of chronic CNO-mediated hM3Dq DREADD activation on expression levels of the neuronal activity marker p-ERK/ERK, we fed 1 mg/kg CNO or vehicle to CamKIIα-tTA::TetO-hM3Dq bigenic mouse pups once daily from P2 to P7 and sacrificed them 15 min post-feeding on P7. Tissue samples were dissected and stored at −80℃, and then homogenized in Radioimmunoprecipitation assay (RIPA) buffer (10 mM Tris-Cl (pH 8.0), 1 mM EDTA, 0.5 mM EGTA, 1% Triton X-100, 0.1% sodium deoxycholate, 0.1% SDS, 140 mM NaCl) using a Dounce homogenizer. The lysis buffer contained protease and phosphatase inhibitors (Sigma- Aldrich, United States). Following the estimation of protein concentration using the Quantipro BCA assay kit

(Sigma-Alrich, United States), equal amounts of lysate were resolved on a 10% sodium dodecyl sulfate polyacrylamide gel and then transferred onto polyvinylidene fluoride membranes. Blots were blocked in 5% milk dissolved in TBST for 1 hr, and subsequently incubated overnight with respective primary antibodies that is rabbit anti-HA (1:1500 in 5% milk, Cat. No. H6908, Sigma-Aldrich, United States), rabbit anti-c-Fos (1:1000 in 5% milk, Cat. No. 2250, Cell Signalling Technology, United States), rabbit anti-actin (1: 10,000 in 5% BSA, Cat. No. AC026, Abclonal Technology, United States), rabbit anti-p-ERK1/2 (Thr202/Tyr204) (1:1000 in 5% BSA, Cat. No. 9101, Cell Signalling Technology, United States), or rabbit anti-ERK1/2 (1:1000 in 5% BSA, Cat. No. 9102, Cell Signalling Technology, United States). Following subsequent washes, blots were exposed to HRP conjugated goat anti-rabbit secondary antibody (1:6000, Cat. No. AS014, Abclonal Technology, United States) for 1 hr. Signal was visualized using a GE Amersham Imager 600 (GE life sciences, United States) with a western blotting detection kit (WesternBright ECL, Advansta, United States). Densitometric quantitative analysis was performed using ImageJ software.

## Immunofluorescence

HA-tagged hM3Dq DREADD expression in the hippocampus and cortex of CamKIIα-tTA::TetO-hM3Dq bigenic mice (P7) was visualized using immunofluorescent staining for the HA epitope. Pups single-positive for either CamKIIα-tTA or TetO-hM3Dq were used as the genotype-controls. Double immunofluorescence stainings were performed on brain sections derived from adult CamKIIα-tTA:: TetO-hM3Dq bigenic mice. Mice were sacrificed by transcardial perfusion with 4% paraformaldehyde, and 40 µm thick serial coronal sections were obtained using a vibratome (Leica, Germany). Following a permeabilization step at room temperature in phosphate-buffered saline with 0.4% Triton X-100 (PBSTx) for 1 hr, the sections were then incubated in the blocking solution [1% Bovine Serum Albumin (Roche, Cat. No. 9048-49-1), 5% Normal Goat Serum (Thermoscientific, Cat. No. PI-31873) in 0.4% PBSTx] at room temperature for 1 hr. The sections were incubated with primary antibody, rabbit anti-HA (1:250; Rockland, Cat. No. 600-401-384, USA). For double-label immunofluorescence experiments, to examine the co-localization of HA-tagged hM3Dq DREADD with markers of excitatory neurons, inhibitory neurons, and glial cells, tissue sections were exposed to the following antibody cocktails: rat anti-HA (1:200, Roche diagnostics, Cat. No. 10145700) with rabbit anti-CamKIIα (1:200, Santa Cruz, Cat. No. sc-12886-R), or rabbit anti-GABA (1:200, Sigma, Cat. No. A2052), or rabbit anti-GFAP (1:500, Chemicon, Cat. No. AB5804) for 4 days at 4°C. Following sequential washes with 0.4% PBSTx, the sections were incubated with the secondary antibody, goat anti-rabbit IgG conjugated to Alexa Fluor 568 (1:500; Invitrogen, Cat. No. A-11011, USA) and goat anti-rat IgG conjugated to Alexa Fluor 488 (1:500; Invitrogen, Cat. No. A-21212, USA), for 2 hrs at room temperature, followed by washes with 0.4% PBSTx. Sections were mounted on to slides using Vectashield Antifade Mounting Medium with DAPI (Vector, H-1200, USA) and images were visualized on a LSM5 exciter confocal microscope (Zeiss, Germany).

## Experimental paradigms
### Postnatal treatment

CamKIIα-tTA::TetO-hM3Dq bigenic males were mated with CamKIIα-tTA::TetO-hM3Dq bigenic females. CamKIIα-tTA::TetO-hM3Dq bigenic females were housed as dyads, prior to single-housing commencing one to two days before parturition. Dams were provided with equal amount of paper shredding as nesting material. The litters were randomly assigned to postnatal CNO (PNCNO) treatment and vehicle-treated control groups. To determine the influence of chronic CNO-mediated hM3Dq DREADD activation of CamKIIα-positive forebrain excitatory neurons during the early postnatal window, CamKIIα-tTA::TetO-hM3Dq bigenic mouse pups were fed either CNO (1 mg/kg) or vehicle for thirteen days from postnatal Day 2 (P2) to postnatal Day 14 (P14) using a micropipette (0.5–10 µl; Cat. No. 3123000020; Eppendorf, Germany). Mouse pups for each postnatal treatment experiment were selected from at least six different dams to minimize any litter-related bias (6–8 pups/litter, one litter/cage). The weight of the pups was measured daily during the course of treatment. Reflex behavior was assayed at postnatal Days 9 and 12, and eye-opening time was observed. The pups were weaned between postnatal Days 25–30, and then housed in identical group-housing conditions till adulthood (3 months onwards) following which a battery of behavioral assays was performed in order to test for persistent alterations in behavior. Adult male and female mice were

assayed for anxiety- and depressive-like behavior, and male mice were assessed for effects on stereotypy and sensorimotor gating.

In order to control for possible off-target effects of postnatal administration of the hM3Dq DREADD agonist CNO, two different experimental paradigms were performed using mice that did not express the hM3Dq DREADD, and one experimental paradigm involved the use of CamKIIα-tTA::TetO-hM3Dq bigenic mice that were administered the alternative DREADD agonist compound 21 (C21). The first experiment involved single-positive mouse pups that were positive for either CamKIIα-tTA or TetO-hM3Dq (genotype-control). The second control experiment involved the use of mouse pups from the background strain C57BL/6J. In both the experiments, the litters were randomly assigned to vehicle (Veh) or postnatal CNO (PNCNO) treatment groups. The pups were weaned between postnatal Days 25–30, and then housed in identical group-housing conditions (3–5 mice/cage) till adulthood (3 months onwards) following which behavioral analysis was performed. Adult male mice from both control experiments were assayed for anxiety- and depressive-like behavior.

Further in a third experiment, we utilized an alternate DREADD agonist, compound 21 and addressed effects in CamKIIα-tTA::TetO-hM3Dq bigenic mice. The litters were randomly assigned to postnatal C21 (PNC21) treatment and vehicle-treated control groups. To determine the influence of chronic C21-mediated hM3Dq DREADD activation of CamKIIα-positive forebrain excitatory neurons during the early postnatal window, CamKIIα-tTA::TetO-hM3Dq bigenic mouse pups were fed either C21 (1 mg/kg) or vehicle for thirteen days from postnatal Day 2 (P2) to postnatal Day 14 (P14) using a micropipette. The paradigm was identical to the PNCNO-treatment paradigm with the difference being that we used an alternative DREADD agonist C21.

## Juvenile treatment

CamKIIα-tTA::TetO-hM3Dq bigenic male mice were weaned between postnatal Days 26–28, and then randomly assigned to vehicle (Veh) or juvenile CNO (JCNO) treatment groups (3–5 animals/cage). To determine the influence of chronic CNO-mediated hM3Dq DREADD activation of CamKIIα-positive forebrain excitatory neurons in the juvenile window, CamKIIα-tTA::TetO-hM3Dq bigenic male mice were fed either CNO (1 mg/kg) or vehicle for thirteen days from postnatal Day 28 (P28) to postnatal Day 40 (P40) using a micropipette (0.5–10 µl; Cat. No. 3123000020; Eppendorf, Germany). Following JCNO treatment, mice were housed in identical group-housing conditions, and tested for anxiety and depressive-like behavior from the age of 3 months onwards, followed by assessment of sensorimotor gating.

## Adult treatment

Adult CamKIIα-tTA::TetO-hM3Dq bigenic male mice (3–4 months of age) were randomly assigned to either vehicle or adult CNO (ACNO) treatment groups (3–5 mice/cage). Mice were taken from at least four different litters to minimize any litter-related bias. CamKIIα-tTA::TetO-hM3Dq bigenic male mice received either vehicle (0.9% saline) or CNO (1 mg/kg CNO in 0.9% saline) for thirteen days once daily via intraperitoneal injections. The influence of CNO-mediated hM3Dq DREADD activation on anxiety-like behavior was assessed both during and soon after cessation of the treatment paradigm. Anxiety-like behavior was assessed using the open field test (OFT) performed on Day 8 of treatment paradigm, the elevated plus maze (EPM) test performed on Day 15, and the light-dark avoidance test carried out on Day 22, with the treatment paradigm being carried out from Day 1 to Day 13. To assay for persistent alterations in emotional behavior following chronic CNO-mediated hM3Dq DREADD activation, CamKIIα-tTA::TetO-hM3Dq bigenic male mice treated chronically with either vehicle/CNO as described above, were given a washout period of three months and then were assayed for anxiety- and depressive-like behavior, followed by the PPI test for sensorimotor gating.

## Behavioral tests

### Reflex behaviors

To assess the influence of chronic CNO-mediated hM3Dq DREADD activation of CamKIIα-positive forebrain excitatory neurons in the early postnatal window on the emergence of reflex behaviors, vehicle and PNCNO-treated CamKIIα-tTA::TetO-hM3Dq bigenic mouse pups were assayed for

surface righting and negative geotaxis on postnatal Days 9 and 12. For surface righting behavior, pups were placed in the home cage upside down, and the time taken to stand on all four paws was noted. For negative geotaxis, the pups were placed on a slanted platform 30° to horizontal, facing the ground and the time taken to face upwards (180° from the initial position) was noted. In addition, the eye-opening time for all mouse pups across postnatal treatment groups was noted.

## Anxiety-like behavior
In order to determine the influence of postnatal, juvenile, and adult treatments on anxiety-like behaviors, CamKIIα-tTA::TetO-hM3Dq bigenic mice from the PNCNO (males and females), JCNO (males), and ACNO (males) treatment groups with their respective vehicle-treated controls were subjected to the open field test (OFT), elevated plus maze test (EPM), and light-dark (LD) avoidance test. Further, PNCNO-treated genotype-control and C57BL/6J male mice, with their individual vehicle-treated control groups were also assessed on the OFT, EPM and LD avoidance test.

## Open field test
Mice were released into one corner of the open field box (40 cm x 40 cm x 40 cm), the center area (20 cm x 20 cm) of which is considered to be anxiogenic. They were released in different corners of the arena in each trial to remove any side bias. Behavior was recorded for 10 min using an overhead analog camera. The captured video was digitized at 25 fps using an analog to digital converter (Startech, UK) and then tracked using an automated online behavioral analysis software (Ethovision XT 11; Noldus, Netherlands). The total distance traveled, percent distance traveled in the center, percent time spent in the center, and number of entries to the center of the arena were calculated.

## Elevated plus maze
The elevated plus maze consisted of a plus-shaped platform with two closed and two open arms (30 cm x 5 cm each) that was elevated 50 cm above the ground. The height of the closed arm walls was 15 cm. The mice were introduced to the arena facing the open arm and behavior was recorded for 10 min using an overhead camera. Following digitization at 25 fps, the behavior was tracked online using the automated behavior analysis platform Ethovision XT 11. The total distance traveled, percent distance traveled in the open arms, percent time spent in the open arms, and number of entries to the open arms were calculated to assess anxiety-like behavior.

## Light-dark avoidance test
The light-dark box consisted of a rectangular box with a light chamber (25 x 25 cm) and a dark chamber (15 x 25 cm) which were connected via a passage (10 x 10 cm). The mice were released into the behavioral arena facing the light box following which behavior was recorded for 10 min using an overhead camera, digitized at 25 fps. The time spent and number entries in the light box were then manually assessed by an experimenter blind to the experimental treatment groups.

## Despair-like behavior
In order to test the effect of postnatal, juvenile, and adult treatments on despair-like behavior, forced swim test (FST) was performed with the PNCNO (males and females), JCNO (males), and ACNO (males) treatment groups. Further, FST was also performed on vehicle/PNCNO-treated genotype-control and C57BL/6J male mice.

## Forced swim test
The forced swim test was performed in a transparent cylindrical chamber (50 cm height, outer diameter: 15 cm, inner diameter: 14 cm) filled with 25°C water to a height of 30 cm from the base. The mice were released into the water and behavior was recorded for 6 min using a side-mounted webcam (Logitech, Switzerland). The time spent immobile was calculated for a duration of 5 min, with the first minute discarded from the analysis by an experimenter blind to the experimental treatment groups.

### Sensorimotor gating

To determine the influence of chronic CNO-mediated hM3Dq DREADD activation of CamKIIα-positive forebrain excitatory neurons during the early postnatal window on sensorimotor gating behavior, vehicle and PNCNO-treated CamKIIα-tTA::TetO-hM3Dq bigenic male mice were assayed on the PPI test performed using a startle and fear conditioning apparatus (Panlab, Spain). In addition, CamKIIα-tTA::TetO-hM3Dq bigenic mice treated with vehicle or CNO during the juvenile (JCNO) or adult (ACNO) window were also assayed for PPI to observe the long-term influence of chronic CNO-mediated hM3Dq DREADD activation of CamKIIα-positive forebrain excitatory neurons during different time epochs on sensorimotor gating behavior. The apparatus comprised of a soundproof chamber with metal grid flooring, a strain gauge coupled to load cells to transduce rapid load change during startle behavior, a load cell amplifier, and a control/interface unit connected to the computer. The load cell was calibrated prior to behavioral testing using a 20 g standard weight by setting the load cell amplifier in DC mode at a gain of 1000. The mouse was placed inside a restrainer to limit spatial location with respect to sound and habituated to the apparatus for four days, followed by habituation for three days with 65 dB background white noise. On the test day, the load cell amplifier was set in AC mode at a gain of 5000. Packwin software (Panlab, Spain) was used to program the protocol and acquire the data. A digital gain of 8 was applied while acquiring the data. The mouse was first habituated to the box for 5 min with 65 dB background white noise which was followed by the first block in which ten tone pulses (120 dB, 1 s) were presented to measure basal startle response. In the second block, the mouse was presented with either only tone (120 dB, 1 s; x10) or a 100 ms prepulse which was either +4 dB, +8 dB or +16 dB higher than the background noise (69/73/81 dB, 1 s; x5) which co-terminated with a 1 s, 120 dB tone. The percent PPI was calculated using the following formula: Percent PPI = 100 × (average startle response with only tone − average startle response with the prepulse) / average startle response with only tone.

### Marble burial

To determine the influence of chronic CNO-mediated hM3Dq DREADD activation of CamKIIα-positive forebrain excitatory neurons during the early postnatal window on repetitive behavior (stereotypy), vehicle and PNCNO-treated CamKIIα-tTA::TetO-hM3Dq bigenic male mice were tested on the marble burial test. The test mice were placed in a cage (30 x 15 cm) where twelve marbles were distributed at an equal interval of ~6 cm on a 4 cm deep bedding material (corn cob). The mice were then allowed to explore the arena for 30 min and the number of marbles buried were assessed by three different experimenters, blind to the experimental treatment groups. Marbles with more than two-thirds of the marble not visible above the surface were considered to be buried. The number of marbles buried per mouse was expressed as the average of the values observed by the three experimenters.

## Neurometabolic analysis in hippocampus and cerebral cortex

To determine the influence of chronic CNO-mediated hM3Dq DREADD activation of CamKIIα-positive forebrain excitatory neurons during the early postnatal window on neurometabolism in the hippocampus and cerebral cortex of vehicle and PNCNO-treated CamKIIα-tTA::TetO-hM3Dq bigenic male mice, the $^{13}$C labeling of brain metabolites were measured in tissue extracts using $^1$H-[$^{13}$C]-NMR spectroscopy following infusion of [1,6-$^{13}$C$_2$]glucose.

### Infusion of [1,6-$^{13}$C$_2$]Glucose

Vehicle and PNCNO-treated adult CamKIIα-tTA::TetO-hM3Dq bigenic male mice were subjected to 6 hr of fasting, and briefly restrained for the cannulation of the tail-vein to infuse $^{13}$C-labeled [1,6-$^{13}$C$_2$]glucose. [1,6-$^{13}$C$_2$]Glucose (ISOTEC, Miamisburg, United States) dissolved in deionized water (0.225 mol/L) was first administered as a bolus followed by an exponentially decreasing infusion rate for 2 min. Blood was withdrawn via retro-orbital bleeding under mild chloroform anesthesia just before the end of the experiment, centrifuged, and the plasma was collected and frozen in liquid N$_2$, and subsequently stored at −80°C. Exactly 6 min following the start of infusion, mice were sacrificed by Focused Micorwave Beam Irradiation (4KW for 0.94 s) in order to instantly arrest neurometabolic activity using the Muromachi Microwave Fixation System (MMW-05, Muromachi, Japan).

The hippocampus and cortex tissues were dissected on ice, snap-frozen in liquid $N_2$, and stored at −80°C until further processing.

## Preparation of brain extract

Metabolites were extracted from brain tissue using a modified protocol described previously (*Patel et al., 2001*). In brief, the frozen hippocampal and cortical tissue samples were homogenized in 3X volume/weight of 0.1 mol/L HCl dissolved in methanol using a motorized homogenizer. [2-$^{13}$C] Glycine (0.2 µmol, 99 atom %, Cambridge Isotopes, Andover, MA, United States) was added to the homogenate as a concentration reference. Following further homogenization, 6X volume/weight of 90% ice-cold ethanol was added and the tissue was homogenized, which was then centrifuged at 16,000 g for 45 min at 4°C. The supernatant was collected and passed through a custom made Chelex column (Biorad, USA). The pH of the extract was adjusted to 7.0, which was followed by lyophilization. The lyophilized powder was dissolved in phosphate-buffered deuterium oxide containing sodium trimethylsilylpropionate (0.25 mmol/L).

## NMR analysis of plasma and brain extract

Blood plasma (100 µl) was mixed with deuterium oxide (450 µl) containing sodium formate (1 µmol/ L) and then passed through a 10 KDa cutoff centrifugal filter (VWR, Radnor, PA, United States). $^1$H-NMR spectra were acquired using a 600 MHz spectrometer (Bruker AVANCE II, Karlsruhe, Germany). Percent $^{13}$C enrichment of glucose-C1 was calculated by dividing the area of the $^{13}$C-coupled satellites by the total area $^1$H area ($^{12}$C + $^{13}$C) observed at 5.2 ppm. $^1$H-[$^{13}$C]-NMR spectra of hippocampal and cortical tissue extracts were recorded as described previously (*Saba et al., 2017*; *de Graaf et al., 2003*; *Bagga et al., 2013*). Briefly, two spin-echo $^1$H-NMR spectra were recorded with an OFF/ON $^{13}$C inversion pulse. Free induction decays (FIDs) were zero-filled, apodized to Lorentzian line broadening, Fourier transformed, and phase-corrected. C-13 edited NMR was obtained by subtracting the sub-spectrum obtained with $^{13}$C inversion pulse from that acquired without inversion. The concentration of metabolites was calculated by using [2-$^{13}$C]glycine as the relative standard. Percentage $^{13}$C enrichment of the desired metabolites was determined as the ratio of the peak areas of the $^1$H-[$^{13}$C]-NMR difference spectrum ($^{13}$C only) to the non-edited spectrum ($^{12}$C + $^{13}$C). This was further corrected for the natural abundance (1.1%) of $^{13}$C.

## Estimation of metabolic rate of glucose oxidation

The metabolic rates of excitatory and inhibitory neurons were calculated using $^{13}$C label trapped into brain metabolites as described previously (*Patel et al., 2005*; *Mishra et al., 2018*). The metabolic rate of glucose oxidation by glutamatergic neurons ($MR_{Glu}$) was calculated using the following formula:

$$MR_{Glu} = 0.5 \times (1/10) \times (1/Glc_{C1}) \times \{0.82[Glu](Glu_{C4} + 2Glu_{C3}) + 0.42[Asp](2Asp_{C3})\}$$

where $Glu_{Ci}$ and $Asp_{Ci}$ are percentage $^{13}$C enrichment of glutamate and aspartate, respectively, at the 'i'th carbon during 6 min infusion, $Glc_{C1}$ is the percent labeling of [1,6-$^{13}$C$_2$]glucose in blood plasma, and [Glu] and [Asp] are the concentrations of glutamate and aspartate, respectively. The metabolic rate of glucose oxidation by GABAergic neurons, $MR_{GABA}$ was calculated by the following formula:

$$MR_{GABA} = 0.5 \times (1/10) \times (1/Glc_{C1}) \times \{0.02[Glu](Glu_{C4} + 2Glu_{C3}) \\ + [GABA](GABA_{C2} + 2GABA_{C4}) + 0.42[Asp](2Asp_{c3})\}$$

$GABA_{Ci}$ is the percentage labeling of GABA at carbon 'i' and [GABA] is the concentration of GABA. The total metabolic rate of glucose oxidation was calculated using the following formula:

$$MR_{Total} = 0.5 \times (1/10) \times (1/Glc_{C1}) \times \{[Glu](Glu_{C4} + 2Glu_{C3}) + [GABA](GABA_{C2} \\ + 2GABA_{C4}) + [Asp](2Asp_{C3}) + [Gln(Gln_{C4})]\}$$

$Gln_{C4}$ refers to the percentage $^{13}$C labeling of glutamine.

## Quantitative PCR

To determine the influence of chronic CNO-mediated hM3Dq DREADD activation of CamKIIα-positive forebrain excitatory neurons during the early postnatal window on persistent changes in gene expression within the hippocampus, hippocampi derived from vehicle and PNCNO-treated CamKIIα-tTA::TetO-hM3Dq bigenic adult male mice were subjected to qPCR analysis. Vehicle and PNCNO-treated adult CamKIIα-tTA::TetO-hM3Dq bigenic mice were anesthetized by $CO_2$ inhalation and sacrificed by rapid decapitation. The hippocampi were then dissected in ice-cold PBS, snap-frozen in liquid $N_2$, and stored at −80°C. RNA extraction was performed using Trizol (TRI reagent, Sigma-Aldrich, USA). The RNA was quantified using a Nanodrop (Thermo Scientific, USA) spectrophotometer followed by reverse transcription reaction to produce cDNA using PrimeScript RT Reagent Kit (Takara, Clonetech, Japan). Specific primers against the genes of interest (*Figure 6— source data 1*) were designed and qPCR was performed to amplify the genes of interest using the CF96X Real Time System (BioRad, USA). The qPCR data were analyzed using the ΔΔCt method as described previously (*Bookout and Mangelsdorf, 2003*). Ct value for a particular gene was normalized to the endogenous housekeeping gene GAPDH (Glyceraldehyde 3-phosphate dehydrogenase), which was unchanged across treatment groups.

## c-Fos immunohistochemistry and cell counting analysis

To determine the influence of chronic CNO-mediated hM3Dq DREADD activation of CamKIIα-positive forebrain excitatory neurons during the early postnatal window on persistent changes in neuronal activity within the hippocampus, brain sections were subjected to c-Fos immunohistochemistry and cell counting analysis. Vehicle and PNCNO-treated CamKIIα-tTA::TetO-hM3Dq bigenic adult male mice that were naïve for behavioral testing, were sacrificed by transcardial perfusion with 4% paraformaldehyde. Coronal sections of 40 μm thickness were obtained using the vibratome (Leica, Germany). Sections were then blocked at room temperature for 2 hr in 10% horse serum with 0.3% TritonX-100 (made in 0.1M Phosphate buffer) following which they were incubated with rabbit anti-c-Fos antibody (1:1000, Cat no. 2250, Cell Signalling Technology, United States) for 2 days at 4°C. Subsequently, they were subjected to incubation with the secondary antibody (biotinylated goat anti-rabbit, 1:500, Cat no. BA9400, Vector Labs, United States) for 2 hr at room temperature. Signal was amplified using an Avidin-biotin complex based system (Vector lab, Vectastain ABC kit Elite PK1600, United States) and then visualized using Diaminobenzidine tetrahydrochloride substrate (Cat no. D5905, Sigma-Aldrich, United States).

An experimenter blind to the treatment groups carried out cell counting of c-Fos immunopositive cells in the hippocampal subfields namely the CA1, CA3, and dentate gyrus (DG) using a brightfield microscope (Zeiss Axioskop two plus, Germany) at a magnification of 200X. Eight sections, separated by a periodicity of 200 μm, spanning the rostrocaudal extent of the hippocampus (four dorsal and four ventral) were selected from each mouse. Results are expressed as the number of c-Fos-positive cells per section for each hippocampal subfield.

## Electrophysiological studies

In order to determine the influence of CNO-mediated hM3Dq DREADD activation on spiking activity, drug-naïve CamKIIα-tTA::TetO-hM3Dq bigenic mouse pups were sacrificed on postnatal Day 7 following which current clamp recordings were performed with CNO bath application. To observe the effects of CNO-mediated chronic postnatal hM3Dq DREADD activation of CamKIIα-positive excitatory neurons on hippocampal neurotransmission, CamKIIα-tTA::TetO-hM3Dq bigenic mice pups were treated once daily with CNO or vehicle from postnatal Day 2–7 (P2–7) and sacrificed on P7 for whole-cell patch clamp recording in aCSF. To determine the influence of chronic CNO-mediated hM3Dq DREADD activation of CamKIIα-positive forebrain excitatory neurons during the early postnatal window (P2–14) on hippocampal neurotransmission that persists into adulthood (3–4 months), whole-cell patch clamp recording was performed on acute hippocampal slices derived from PNCNO and vehicle-treated adult CamKIIα-tTA::TetO-hM3Dq bigenic male mice.

### Preparation of hippocampal slices

Mice were sacrificed by cervical dislocation in accordance with the guidelines of the Jawaharlal Nehru Centre for Advanced Scientific Research (JNCASR, Bengaluru) Institutional Animal Ethics

Committee. The mice were quickly decapitated following which the brain was transferred to ice-cold sucrose cutting solution (189 mM Sucrose, 10 mM D-glucose, 26 mM NaHCO$_3$, 3 mM KCl, 10 mM MgSO$_4$.7H$_2$O, 1.25 mM NaH$_2$PO$_4$, and 0.1 mM CaCl$_2$), bubbled continuously with 95% O$_2$ and 5% CO$_2$. The brain was then dissected in a petri dish containing an ice-cold cutting solution, the cerebellum was removed, and glued onto a brain holder before placing in a container filled with ice-cold cutting solution. Horizontal sections (300 μm) were obtained using a vibrating microtome (Leica, VT-1200, Germany) and transferred to a petri dish containing aCSF (124 mM NaCl, 3 mM KCl, 1 mM MgSO$_4$.7H$_2$O, 1.25 mM NaH$_2$PO$_4$, 10 mM D-glucose, 24 mM NaHCO$_3$, and 2 mM CaCl$_2$) at room temperature. The part of the sections containing the hippocampus was gently dissected out and transferred to a nylon mesh chamber containing aCSF that was continuously bubbled with 95% O$_2$ and 5% CO$_2$ at 37°C. Following a recovery period of 45–60 min at 37°C to ensure a stable electrophysiological baseline response, slices were kept at room temperature and subsequently transferred to the recording chamber as required.

## Recording rig and data acquisition

The slice recording chamber was continuously circulated with aCSF at 1–2 mL min$^{-1}$ using a combination of a peristaltic pump (BT-3001F, longer precision pump Co. Ltd., China) and gravity feed. The aCSF was pre-heated to 34°C using a single channel temperature controller (Cat. No. TC-324C, Warner Instruments, USA) with a feedback temperature control provided through a thermistor submerged in the recording chamber. In addition, an Ag/AgCl wire was submerged in the recording chamber and was used for referencing. The recording electrodes were held at approximately 45° to the vertical and were controlled using a micromanipulator (Cat. No. 1U RACK; Scientifica, UK). The neurons in the slice were visualized using an upright microscope (Slicescope pro 6000 Scientifica, UK) with specialized optics to visualize deep tissue. The recording set up was enclosed in a Faraday cage and placed on an anti-vibration table (Cat. No. 63 P-541; TMC, USA). The electrical noise was eliminated by grounding all electrical connections to a single ground point in the amplifier.

## Recording electrodes

Borosilicate glass capillaries (Cat. No. 30–0044/GC120 F-10; Harvard apparatus, UK) were used to pull recording electrodes using a horizontal micropipette puller (Cat. No. P97, Sutter Instruments Co., USA). Intracellular patch electrodes (6–8 MΩ) used to record spiking activity, intrinsic properties, spontaneous postsynaptic currents (sPSC), spontaneous excitatory postsynaptic currents (sEPSC), and miniature excitatory postsynaptic currents (mEPSC) were filled with potassium gluconate (KGlu) internal solution (130 mM KGlu, 20 mM KCl, 10 mM HEPES free acid, 0.2 mM EGTA, 0.3 mM GTP-Na salt, 4 mM ATP-Mg salt; osmolarity adjusted to 280–310 mOsm). Intracellular patch electrodes (6–8 MΩ) used to record spontaneous inhibitory postsynaptic currents (sIPSC) and miniature inhibitory postsynaptic currents (mIPSC) were filled with cesium chloride (CsCl) internal solution (120 mM CsCl, 10 mM HEPES free acid, 10 mM EGTA, 5 mM QX314-Br, 4 mM ATP disodium salt, 0.3 mM GTP disodium salt, 4 mM MgCl$_2$, osmolality adjusted to 280–310 mOsm). The microelectrodes were mounted to an electrode holder (Scientifica, UK), which was further mounted to a headstage connected to the amplifier. All signals were amplified using a Multiclamp-700B (Molecular Devices, USA).

## Whole-cell patch clamp recording

Somata of CA1 pyramidal neurons were patched in order to perform whole-cell patch clamp recording. A small positive pressure was applied to the patch pipette filled with appropriate intracellular recording solution using a mouth-operated syringe which was attached to the pipette holder using air-tight tubing. After setting current and voltage offset to zero, the resistance of the electrode was noted by applying a test voltage step and measuring the resulting current according to Ohm's law. When the pipette tip touched the cell surface which could be confirmed both by the appearance of 'dimple shape' under the microscope and a change in resistance, the positive pressure was released. A subsequent negative pressure was applied by a gentle suction leading the pipette to form a tight seal with the somata, indicated by a gigaohm-seal (>1 GΩ). The slow and fast capacitance were offset followed by the rupturing of the cell membrane by applying gentle suction. The cells with membrane potential more depolarized than −55 mV in adulthood and −45 mV at P7 were not

considered. In addition, only cells having a series resistance in the range of 5–25 MΩ during the course of recording were considered.

Recording of spiking activity of CA1 pyramidal neurons following CNO bath administration at P7 was performed by holding cells in a current clamp mode. The identity of CA1 pyramidal neurons was confirmed qualitatively using the shape of action potential (characterized by the presence of an after-depolarization potential) by injecting a current of up to 2 nA for 2 ms through the patch electrode. A 5-min baseline was recorded followed by bath application of 1 μM for 2 minutes. The slices were then washed in aCSF and spiking activity was recorded. In order to measure intrinsic membrane properties and input-output characteristics, both at P7 and adulthood, a 500 ms, 7-step hyperpolarizing or depolarizing current (ranging from −100 pA to 180 pA) was injected through the patch electrode at an inter-sweep interval of 10 s.

Spontaneous postsynaptic currents (sPSCs) were measured by holding the cell in voltage-clamp mode at −70 mV. sEPSCs were recorded in the presence of 10 μM bicuculline and mEPSCs were recorded in the presence of 1 μM TTX + 10 μM bicuculline, while cells were held in a voltage-clamp mode at −70 mV. Further, sIPSCs were recorded in the presence of 10 μM CNQX + 100 μM AP5 and mIPSCs were recorded in the presence of 1 μM TTX + 10 μM CNQX + 100 μM AP5 in a voltage-clamp mode at −70 mV.

## Analysis of electrophysiology data

Whole-cell patch clamp data were analyzed offline using the Clampfit 10.5 (Molecular Devices, USA) and Mini Analysis program (Synaptosoft Inc, USA). Voltage deflection traces obtained from a 100 pA hyperpolarizing current pulse were used to calculate different intrinsic membrane properties as follows. The input resistance was calculated by applying Ohm's law, R = V/I. Where V = stable state voltage and I = injected current (100 pA). The voltage trace was fitted to a single exponential $Ae^{-t/\tau}$ and the decay constant was measured in order to calculate the membrane time constant ($\tau_m$). Sag voltage was calculated by subtracting the steady-state voltage from the peak negative-going voltage. The accommodation index was calculated as the ratio of the maximum inter-spike interval (including the interval from the last spike to the end of the current injection) to the first inter-spike interval. The number of spikes fired in response to an increasing amplitude of current injection (0–180 pA) was calculated to generate the input-output curve. Amplitude and interevent intervals for all spontaneous currents were calculated by a semi-automated event-detection algorithm using the Mini Analysis program. At least 200 events from identical temporal bins were counted.

## Statistics

All experiments had two treatment groups and were subjected to a two-tailed, unpaired Student's t-test using GraphPad Prism (Graphpad Software Inc, USA). One-sample Kolmogorov-Smirnov test was performed to confirm normality. All graphs were plotted using GraphPad Prism (Graphpad Software Inc, USA). Data are expressed as mean ± standard error of the mean (S.E.M) and statistical significance was set at $p < 0.05$. To account for type I errors, the qPCR data were further subjected to the two-stage linear step-up procedure of Benjamini, Krieger and Yekutieli method to calculate false discovery rate (FDR) at 5%. Vehicle and PNCNO-treatment groups were subjected to linear regression followed by ANCOVA in order to compare input-output curves and statistical significance was set at $p < 0.05$. For the analysis of spontaneous current data, amplitudes and interevent intervals of events recorded from vehicle or PNCNO-treatment groups were converted to corresponding cumulative probability distributions and then subjected to Kolmogorov-Smirnov two-sample comparison. Statistical significance was set at $p < 0.001$.

## Acknowledgements

We are grateful to Prof. Rishikesh Narayan (Molecular Biophysics Unit, Indian Institute of Science, Bangalore) for his valuable inputs on the manuscript. We thank all members of the Vaidya and Clement Lab for their technical help. We thank Monalisa Ghosh and Manish Biyani from the Patel Lab for their technical support during NMR experiments. We thank TIFR animal Facility personnel, Dr. Sachin Atole and Ms. KV Boby for technical support.

# Additional information

## Competing interests

Vidita A Vaidya: Reviewing editor, *eLife*. The other authors declare that no competing interests exist.

## Funding

| Funder | Grant reference number | Author |
|---|---|---|
| Tata Institute of Fundamental Research | RTI4003 | Vidita A Vaidya |

The funders had no role in study design, data collection and interpretation, or the decision to submit the work for publication.

## Author contributions

Sthitapranjya Pati, Conceptualization, Data curation, Formal analysis, Investigation, Methodology, Writing - original draft, Writing - review and editing; Kamal Saba, Formal analysis, Investigation, Methodology; Sonali S Salvi, Formal analysis, Investigation; Praachi Tiwari, Data curation, Formal analysis, Investigation, Methodology, Writing - review and editing; Pratik R Chaudhari, Vijaya Verma, Shital Suryavanshi, Investigation, Methodology; Sourish Mukhopadhyay, Darshana Kapri, Formal analysis, Investigation, Methodology, Data acquisition; James P Clement, Anant B Patel, Supervision, Investigation, Methodology; Vidita A Vaidya, Conceptualization, Formal analysis, Supervision, Funding acquisition, Writing - original draft, Writing - review and editing

## Author ORCIDs

Sthitapranjya Pati (iD) https://orcid.org/0000-0001-8598-4376
Kamal Saba (iD) https://orcid.org/0000-0003-1729-9961
Vidita A Vaidya (iD) https://orcid.org/0000-0002-3907-8580

## Ethics

Animal experimentation: All animal experiments were carried out in strict accordance with the guideline of the Committee for the Purpose of Control and Supervision of Experiments on Animals (CPCSEA), Government of India. Experiments and use of animals were approved by the institutional ethics committees of the Tata Institute of Fundamental Research, Mumbai, India (TIFR/IAEC/2017-2; 56/GO/ReBi/S/99/CPCSEA); Jawaharlal Nehru Centre for Advanced Scientific Research, Bengaluru, India ( JPC001, JPC005; 201/GO/Re/S/2000/CPCSEA); and Centre for Cellular and Molecular Biology, Hyderabad, India (IAEC 32/2018; 20/GO/RBi/S/99/CPCSEA). Care was taken across all experiments to minimize animal suffering and restrict the number of animals used.

## Decision letter and Author response

Decision letter https://doi.org/10.7554/eLife.56171.sa1
Author response https://doi.org/10.7554/eLife.56171.sa2

# Additional files

## Supplementary files

• Source data 1. Summary of statistical analysis.
• Transparent reporting form

## Data availability

All data generated or analysed during are included in the manuscript and supporting files. Source data files have been provided for Figure 2 and Figure 3.

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
