## [Decision Letter]

Thank you for submitting your article "Chronic postnatal chemogenetic activation of forebrain excitatory neurons evokes persistent changes in mood behavior" for consideration by *eLife*. Your article has been reviewed by two peer reviewers, and the evaluation has been overseen by a Reviewing Editor and Kate Wassum as the Senior Editor The following individuals involved in review of your submission have agreed to reveal their identity: Jessica Bolton (Reviewer #1); Amelia J Eisch (Reviewer #3).

The reviewers have discussed the reviews with one another and the Reviewing Editor has drafted this decision to help you prepare a revised submission.

This manuscript represents an interesting and thorough exploration of whether chronic postnatal chemogenetic activation of forebrain excitatory neurons can "phenocopy" the effects of early-life adversity and provoke long-term changes in emotional function. The authors demonstrate that chronic Gq activation in the forebrain can cause long-term changes in anxiety-like behavior, depressive-like behavior, and pre-pulse inhibition. We noted considerable rigor in the experimental design.

The reviewers and editors agree that your paper is potentially suitable for *eLife*. However, they noted several concerns that should be addressed.

Essential revisions:

1) It is not sufficient to just confirm that forebrain and hippocampus express the Gq-DREADD; controls where you expect no expression need to be included along with cell type-specificity; e.g., is it expressed in CamKIIα+ neurons specifically and not in interneurons and glia?

2) A 2-way ANOVA with sex as a factor needs to be performed to determine whether the males and females are different from each other in how they respond to Gq activation (an interaction with sex will uncover a lack of a sex difference). In addition, the justification for only using males is weak, and needs to be strengthened in the Discussion. Please include males and females in the same figure (Figure 2) for ease of comparison.

3) The decrease in distance traveled in the OFT provides a caveat to the other measures in the OFT, especially since the % time in center is not different. Please discuss the possibility that this could be an effect on locomotion, not on anxiety.

4) Object-discrimination memory does not depend on the hippocampus; why was a hippocampus-dependent task not used since this region is a focus of the report? Animals with early-life adversity are more likely to be hindered on the object-location memory task, which is more challenging and relies on the hippocampus (see Molet et al., 2016, Hippocampus). Please comment on this limitation of your experiments in the Discussion. More importantly, please do not state that you have seen no changes in "cognition", which is far too broad of a category to claim based on the limited behavioral tasks that were tested.

5) The decrease in total distance traveled in the OFT after PNCNO is similar to what is seen the OFT of bigenic mice with PNCNO, so this effect is likely to be due to CNO itself, rather than Gq-DREADD activation. This result needs to be mentioned as an experimental caveat in the Discussion.

6) For more direct translational implications, an obvious extension would be to suggest using Gi-DREADDs in the context of an early-adversity model in order to prevent changes.

7) How exactly was the CNO orally administered from P2-14? Gavage? Did the pups lick the CNO? Please give a strong rationale for the ways that were utilized to ensure that each pup received the exact same dose. Was the CNO dissolved first in DMSO, as is typically done because it is not very water-soluble?

8) The way the task was setup lends itself to a pronounced ceiling effects – it sounds too easy. It is no wonder you didn't see any difference, because everyone is probably at ceiling. Why did mice explore the familiar objects again for 5 minutes on the 5th day, and then why was one familiar object replaced with the novel object? And how can a novel object be inserted in the middle of the test without fully disturbing the mice, and clearly indicating which was the novel object? Typically, the 5th day would be one familiar object and one novel object, in order to assess long-term (24-hour) memory of the familiar object. Please clarify and discuss this.

9) The authors' conclusion of a lack of off-target CNO effects is not definitive without direct comparison of behavioral results among at least three groups (CNO-hM3Dq, CNO-control, and Veh-hM3Dq). If this is not possible due to their breeding strategy, testing an alternative designer drug without back metabolism (C21 or J compound) is needed. If this is not possible, the conclusions should be modified appropriately and the caveats clearly listed.

10) The authors state chemogenetic stimulation of forebrain excitatory neurons during postnatal life (P2-14), but not during the juvenile or adult periods, increases anxiety-, despair-, and schizophrenia-like behavior in adulthood. However, this statement emerges from comparing groups that received CNO (1 mg/kg) via oral administration and i.p. administration. As the level of CNO into the brain varies by drug delivery route, the comparison of different ages should be made among group of mice that received the same route of administration and dose of CNO. If this is not possible, the conclusions should be modified appropriately and the caveats noted.

11) While acute CNO injection in early postnatal life led to an immediate increase in cellular activity, repeated CNO injection in early life (P2-14) led to decreased cellular activity in adulthood, as well as anxiety- and depressive-like behavior. The authors interpret this as early life chemogenetic forebrain stimulation has negative consequences in adulthood. What about the possibility that the early adverse event (stimulation) and its long-lasting negative effect on mood related behaviors are due to seizures or seizure-related brain activity? Prior work from the Roth and McNamara groups shows forebrain chemogenetic mouse models can have a range of behaviorally-relevant symptoms with various CNO doses (and some effects even in the absence of CNO), including seizures. Please discuss. Furthermore, what evidence do the authors have that the early life chemogenetic forebrain stimulation does not increase cell death in adulthood? If no evidence is available, this is a reasonable discussion point.

12) Please ensure that your Materials and methods includes information on mice/cage in all cases.

13) Please ensure that you include full statistical reporting, including F and t values, degrees of freedom etc. for each statistical result.

[Editors' note: further revisions were suggested prior to acceptance, as described below.]

Thank you for resubmitting your work entitled "Chronic postnatal chemogenetic activation of forebrain excitatory neurons evokes persistent changes in mood behavior" for further consideration by *eLife*. Your revised article has been evaluated by Kate Wassum (Senior Editor) and a Reviewing Editor.

The manuscript has been improved but there are some remaining issues that need to be addressed before acceptance, as outlined below:

Please incorporate the primary statistics into the main manuscript. It is okay to also include the source data document, but all primary results should have corresponding statistical comparisons in the main manuscript so readers do not have to download a separate document to see this information.

---

## [Author Response]

Essential revisions:1) It is not sufficient to just confirm that forebrain and hippocampus express the Gq-DREADD; controls where you expect no expression need to be included along with cell type-specificity; e.g., is it expressed in CamKIIα+ neurons specifically and not in interneurons and glia?

We concur with the reviewer that this would be an important additional control, and so we performed double immunofluorescence staining to delineate the cell-type specificity for the expression of the HA-tagged hM3Dq-DREADD. This data has now been included as a new supplementary figure (Figure 1—figure supplement 1), and we have modified our Materials and methods and Results section to reflect these new data. Our immunofluorescence experiments indicate that the HA-tagged hM3Dq-DREADD selectively co-localizes with CamKIIα-positive excitatory neurons in the forebrain, with high expression noted in hippocampal subfields and the neocortex (Figure 1—figure supplement 1B). High magnification images indicate that expression of the HA-tagged hM3Dq-DREADD is likely to be present on the membrane of CamKIIα-positive excitatory neurons (Figure 1—figure supplement 1B). In contrast, we did not observe any HA-tagged hM3Dq-DREADD immunofluorescence staining on either GABAergic inhibitory neurons or on GFAP-positive astrocytes (Figure 1—figure supplement 1C). In addition, we also noted the absence of HA-tagged hM3Dq-DREADD expression in subcortical regions, including the hypothalamus, pallidum, and periaqueductal gray (Figure 1—figure supplement 1D). This is consistent with prior reports from the Roth lab (Alexander et al., 2009) using the bigenic CamKIIα-tTA::TetO hM3Dq mouse line.

2) A 2-way ANOVA with sex as a factor needs to be performed to determine whether the males and females are different from each other in how they respond to Gq activation (an interaction with sex will uncover a lack of a sex difference). In addition, the justification for only using males is weak, and needs to be strengthened in the Discussion. Please include males and females in the same figure (Figure 2) for ease of comparison.

A four-group experiment that assessed effects on both male and female bigenic CamKIIα-tTA::TetO-hM3Dq mice subjected to excitatory DREADD-mediated chemogenetic activation of forebrain excitatory neurons during postnatal life and compared to vehicle-treated male and female control groups would have allowed for a two-way ANOVA analysis to address whether there are any sexually dimorphic effects of postnatal chemogenetic hM3Dq activation. However, we ran these experiments in adult CamKIIα-tTA::TetO-hM3Dq males and females with a history of PNCNO treatment in distinct cohorts of animals, and at different times. Thus, given the behavioral battery was run for the males and females separately, we cannot now combine the data into one figure, and perform 2-way ANOVA on the combined dataset. We agree that this is a caveat to consider while interpreting the results of our study, and further experiments are required to carefully address the possibility of sexually dimorphic effects on mood-related behavior following excitatory DREADD-mediated chemogenetic activation of forebrain excitatory neurons during postnatal life. We have stated this caveat in our Results and Discussion section.

3) The decrease in distance traveled in the OFT provides a caveat to the other measures in the OFT, especially since the % time in center is not different. Please discuss the possibility that this could be an effect on locomotion, not on anxiety.

We agree that a decrease in total distance travelled in the OFT arena (Figure 1F) in PNCNO-treated adult male mice could indicate a potential impact on locomotion, in addition to effects on anxiety-like behavior. Along with a decline in total distance travelled in the center, we also noted a decrease in percent distance travelled (Figure 1C) and number of entries to the center (Figure 1E) as well, suggestive of enhanced anxiety-like behavioral responses in the PNCNO-treated group. These data along with an increased anxiety-like behavior in the EPM and LD box test, suggest that a history of PNCNO-treatment increases anxiety-like behavior in bigenic CamKIIα-hM3Dq mice. In addition, we did not observe any significant difference in total distance travelled in the EPM. However, we cannot completely rule out possible off-target effects of CNO treatment on locomotion considering the fact that our perturbation was carried out during the critical period of maturation of many forebrain circuits. When we used the alternate DREADD agonist Compound 21 (C21), we did not observe any change in total locomotion in the OFT in PNC21 treated adult male mice, but did observe significant increases in anxiety-like behavior. We have now included a point about this in the Discussion section of our manuscript.

4) Object-discrimination memory does not depend on the hippocampus; why was a hippocampus-dependent task not used since this region is a focus of the report? Animals with early-life adversity are more likely to be hindered on the object-location memory task, which is more challenging and relies on the hippocampus (see Molet et al., 2016, Hippocampus). Please comment on this limitation of your experiments in the Discussion. More importantly, please do not state that you have seen no changes in "cognition", which is far too broad of a category to claim based on the limited behavioral tasks that were tested.

We are in agreement with the reviewer that assessing cognitive behavior with only the novel object recognition (NOR) test does not allow us to reach a conclusion about the effects of chronic postnatal activation of forebrain excitatory neurons on cognition. We also agree with the reviewer that the version of NOR used in our study is not a hippocampal-dependent behavioral task. Given the current limitations in our ability to conduct extensive behavioral testing for cognitive behavior in PNCNO-treated CamKIIα-tTA::TetO-hM3Dq mice, we have chosen to exclude this figure from our manuscript which is predominantly focussed on the impact of chronic postnatal activation of CamKIIα-positive forebrain excitatory neurons on mood-related behaviors. We realise that to draw a conclusion about potential effects on cognitive performance would require a detailed independent study to determine the effects of chronic postnatal activation of CamKIIα-positive forebrain excitatory neurons on a battery of cognitive behavioral tasks.

5) The decrease in total distance traveled in the OFT after PNCNO is similar to what is seen the OFT of bigenic mice with PNCNO, so this effect is likely to be due to CNO itself, rather than Gq-DREADD activation. This result needs to be mentioned as an experimental caveat in the Discussion.

We observed a small, but significant decrease in the total distance travelled in the OFT arena in the genotype control animals (Single-positive for either CamKIIα-tTA or TetO-hM3Dq) with a history of PNCNO-treatment (Figure 2—figure supplement 5F). We did not observe an effect of PNCNO-treatment on the total distance travelled in the OFT arena in the background strain (C57BL/6J). We also do not observe any alterations in total distance travelled in the EPM in either the genotype controls or the background strain with a history of PNCNO-treatment. Further, we have performed experiments with an alternate DREADD agonist, Compound 21 (C21) (now included in the manuscript as Figure 2—figure supplement 7) which when used to chemogenetically activate forebrain excitatory neurons during postnatal life (P2-14) resulted in significant increases in anxiety-like behavior in adulthood on the OFT and EPM. We did not observe any decrease in total locomotion on the OFT or EPM as a consequence of PNC21 treatment. This collectively suggests that chemogenetic activation of forebrain excitatory neurons during postnatal life does not appear to exert any robust effects on adult locomotor behavior. However, we acknowledge that the effect observed in Figure 2—figure supplement 5 does not allow us to completely rule out some off-target effects of postnatal CNO treatment on total locomotion in the OFT. We have now addressed this caveat in our Discussion.

6) For more direct translational implications, an obvious extension would be to suggest using Gi-DREADDs in the context of an early-adversity model in order to prevent changes.

We concur with the reviewer that postnatal activation of Gi-DREADDs in forebrain excitatory neurons would be one of the next logical questions to address. This is particularly relevant in relation to the developmental programming of mood-related behavior by the serotonergic pathway. The excitatory Gq-coupled serotonin receptor 2A (5-HT2A) and inhibitory Gi-coupled serotonin receptor 1A (5-HT1A) have been most widely studied in this regard. The developmental knockout of the 5-HT2A receptor reduces anxiety-like behavior (Weisstaub et al., 2006), whereas knocking out the 5-HT1A receptor leads to increased anxiety (Gross et al., 2002). These effects can be restored by developmentally rescuing the receptor expression only in the neocortex (Gross et al., 2002; Weisstaub et al., 2006). Past studies from our lab involving pharmacological perturbations further confirm and substantiate the hypothesis that a balance between activity mediated by these two serotonergic receptors, coupled to Gq and Gi signalling, could be crucial in long-term programming of mood-related behavior. We are currently performing experiments with Gi DREADDs to extensively examine consequences of enhanced Gi signalling in forebrain excitatory neurons during postnatal life on mood-related behaviour, the scope of which falls under a separate line of study. As suggested by the reviewer, we have included this point in our Discussion.

7) How exactly was the CNO orally administered from P2-14? Gavage? Did the pups lick the CNO? Please give a strong rationale for the ways that were utilized to ensure that each pup received the exact same dose. Was the CNO dissolved first in DMSO, as is typically done because it is not very water-soluble?

For the purpose of feeding, CNO (Tocris, Cat. No. 4936, UK) was directly dissolved in 5% aqueous sucrose solution. We have noted that up to 3 mg/ml CNO is readily soluble in water at room temperature and gentle vortexing. DREADD agonist compound 21 (C21; Tocris, Cat. No. 5548, UK) was dissolved in 10 μl DMSO, and then diluted in 5% aqueous sucrose solution to a 1 ml stock solution. Following this, these solutions were aliquoted and stored at -80℃ before using for oral delivery in postnatal experiments.

The pups were fed by bringing the pipette close to their mouth and dispensing the liquid on their tongue. For this purpose, a 10 μl pipette and appropriate tips were used. The pups readily licked all the solution in about two days after feeding started. Although we have not checked the brain availability of CNO, we do not see a large variance in either the levels of ERK phosphorylation or c-Fos induced following CNO-mediated hM3Dq DREADD activation in both acute (Figure 1E, F) and chronic (Figure 1—figure supplement 1) CNO-treated groups. While we are aware that gavage may ensure lesser variance in brain availability of CNO, we have deliberately avoided gavage in our feeding experiments as it could be stressful for the pups and increase handling time.

8) The way the task was setup lends itself to a pronounced ceiling effects – it sounds too easy. It is no wonder you didn't see any difference, because everyone is probably at ceiling. Why did mice explore the familiar objects again for 5 minutes on the 5th day, and then why was one familiar object replaced with the novel object? And how can a novel object be inserted in the middle of the test without fully disturbing the mice, and clearly indicating which was the novel object? Typically, the 5th day would be one familiar object and one novel object, in order to assess long-term (24-hour) memory of the familiar object. Please clarify and discuss this.

We designed the task to assess short-term object recognition memory. We agree with the reviewer that simply using this experiment to interpret cognitive consequences of PNCNO-treatment would not be ideal. Given the current limitations in our ability to conduct extensive behavioral testing for cognitive behavior in PNCNO-treated CamKIIα-tTA::TetO hM3Dq mice, we have chosen to exclude this figure from our manuscript which is predominantly focussed on the impact of chronic postnatal activation of CamKIIα-positive forebrain excitatory neurons on mood-related behaviors. We thank the reviewers for their feedback, and agree that extensive additional experiments are required prior to drawing a conclusion on the cognitive consequences of PNCNO treatment.

9) The authors' conclusion of a lack of off-target CNO effects is not definitive without direct comparison of behavioral results among at least three groups (CNO-hM3Dq, CNO-control, and Veh-hM3Dq). If this is not possible due to their breeding strategy, testing an alternative designer drug without back metabolism (C21 or J compound) is needed. If this is not possible, the conclusions should be modified appropriately and the caveats clearly listed.

We do agree with the reviewer that we have not performed a direct comparison between the bigenic and genotype/background control groups. This was not possible as animals used in these control experiments belonged to different cohorts and experiments were performed at different time points. We also agree that postnatal treatment with another DREADD agonist other than CNO could serve as an essential control to further rule out nonspecific off-target effects of CNO administration. As per the suggestion of the reviewer, we orally administered the DREADD agonist compound 21 (C21) to bigenic CamKIIα-hM3Dq mice during the first two weeks of postnatal life and assayed for anxiety-like behavior in adulthood on the open field test (OFT) and elevated plus maze test (EPM). This experimental paradigm was identical to the PNCNO-treatment paradigm with the only difference being the administration of C21 instead of CNO. In agreement with the anxiogenic effects observed in adulthood in mice with a history of PNCNO-treatment, we noted that chronic chemogenetic activation of CamKIIα-positive forebrain excitatory neurons using C21 during the early postnatal window results in a long-lasting increase in anxiety-like behavior in adult mice both on the OFT and EPM. We have now included these results in our manuscript (Figure 2—figure supplement 7). We thank the reviewer for their constructive input and believe this has strengthened our study. We have also added this point to our Discussion section.

10) The authors state chemogenetic stimulation of forebrain excitatory neurons during postnatal life (P2-14), but not during the juvenile or adult periods, increases anxiety-, despair-, and schizophrenia-like behavior in adulthood. However, this statement emerges from comparing groups that received CNO (1 mg/kg) via oral administration and i.p. administration. As the level of CNO into the brain varies by drug delivery route, the comparison of different ages should be made among group of mice that received the same route of administration and dose of CNO. If this is not possible, the conclusions should be modified appropriately and the caveats noted.

We chose the drug administration route in mice based on our knowledge of what minimizes handling-associated stress at the particular age of treatment. In our experience, feeding in pups and juveniles is far less stressful than repeated intraperitoneal (i.p.) injections. In unpublished results from our lab, we have compared i.p. injections versus feeding-based drug administration and find that repeated i.p. injections in the postnatal and juvenile window are stressful. Feeding pups or juveniles via a micropipette, minimizes the duration of handling (~5 secs) and does not require restraining the animals in a tight hold. The animals were held in the left hand within the palm and the thumb, index, and middle fingers were gently placed around to stop it from moving. In contrast, we administered chronic CNO to adults through the i.p. route, as gavage would have been more stressful and would require longer restraint in adult mice. However, we do agree with the reviewer that different routes of administration may result in variable levels of brain availability of the DREADD agonist CNO. Thus, it is not possible for us to conclude that the effects observed across lifespan involved equivalent pharmacodynamics of CNO. We would like to point out that i.p. route would normally result in higher plasma levels of the drug as compared to the oral route. Thus, our working estimate is the brain availability of CNO is either equivalent or more in adult mice as compared to the effective concentration in either the pups or juvenile mice. Thus, it is noteworthy that there is no effect on anxiety-like behavior following chronic administration of CNO in adult CamKIIα-hM3Dq bigenic mice. We thank the reviewers for their suggestion to include this point in our Discussion.

11) While acute CNO injection in early postnatal life led to an immediate increase in cellular activity, repeated CNO injection in early life (P2-14) led to decreased cellular activity in adulthood, as well as anxiety- and depressive-like behavior. The authors interpret this as early life chemogenetic forebrain stimulation has negative consequences in adulthood. What about the possibility that the early adverse event (stimulation) and its long-lasting negative effect on mood related behaviors are due to seizures or seizure-related brain activity? Prior work from the Roth and McNamara groups shows forebrain chemogenetic mouse models can have a range of behaviorally-relevant symptoms with various CNO doses (and some effects even in the absence of CNO), including seizures. Please discuss. Furthermore, what evidence do the authors have that the early life chemogenetic forebrain stimulation does not increase cell death in adulthood? If no evidence is available, this is a reasonable discussion point.

The Roth and McNamara groups have reported seizure activity following intraperitoneal administration of CNO in mice expressing the hM3Dq DREADD in CamKIIα-positive forebrain excitatory neurons. However, the effect observed by them is far less penetrant at lower doses (e.g. 1 mg/kg i.p. injection induced seizure epilepticus in about 20-30% mice). In contrast to the effect observed by them, we have not noted any visible signs of seizing on the Racine scale, in either pups or adult mice following administration of CNO. We, however, cannot rule out the possibility of microseizure-like behavior and associated neurophysiological changes in bigenic CamKIIα-hM3Dq mice following oral administration of CNO during the postnatal window. Secondly, recent results from the Marin lab demonstrate that activity of pyramidal neurons can bi-directionally regulate cell-death in the neocortex (Wong et al., 2018). Apoptotic cell death also constitutes one of the key physiological phenomena during the maturation of neocortical microcircuitry, with approximately 25-35% cell death occurring during the first two weeks of postnatal life (Wong and Marin, 2019). We have not assessed the effect of chronic chemogenetic activation of CamKIIα-positive forebrain excitatory neurons during early postnatal window on cell death in the neocortex. Thus, we cannot rule out the possibility of the effect of PNCNO-treatment on cell death, and its subsequent impact on adult neurophysiology and behavior.

Modulation of cell-death could potentially serve as one of the many downstream effectors via which chronic increase in forebrain excitatory tone during postnatal life evokes long-lasting behavioral alterations and this motivates future experiments to delineate the downstream cellular/circuit mechanisms. We have now mentioned this in our Discussion.

12) Please ensure that your Materials and methods includes information on mice/cage in all cases.

For the postnatal feeding experiments, we have used a litter size of 6-8 pups/litter with one litter housed per cage, with each postnatal treatment experiment using pups derived from multiple litters to minimize any litter-related bias. These animals were segregated into 3-5 mice/cage post-weaning and stayed in these throughout the course of behavioral experiments. Barring for the female PNCNO-treatment experiment (Figure 1—figure supplement 3), female pups were culled at the stage of weaning to manage space availability in the animal facility. For all juvenile and adult experiments 3-5 male mice were housed per cage both during CNO treatment and while performing behavioral assays.

Postnatal experiments: 6-8 pups/litter. 1 litter/cage

Juvenile/adult experiment: 3-5 mice/cage.

13) Please ensure that you include full statistical reporting, including F and t values, degrees of freedom etc. for each statistical result.

We have now included a sheet summarizing relevant statistics, including F-values, t-values, and degrees of freedom for two-tailed unpaired t-tests and D-values for two-sample Kolmogorov-Smirnov tests. We have also included the sample size, including number of animals and number of cells as required. In order to improve the readability of our manuscript, we have decided not to include all this information within the results or figure legends of our main manuscript, but have made it available as additional metadata that can be linked to the manuscript.

[Editors' note: further revisions were suggested prior to acceptance, as described below.]

Please incorporate the primary statistics into the main manuscript. It is okay to also include the source data document, but all primary results should have corresponding statistical comparisons in the main manuscript so readers do not have to download a separate document to see this information.

As per the suggestion of the reviewers, we have now incorporated F-values in addition to *p*-values in the Results section our main manuscript (highlighted in blue) for all two-tailed unpaired t-tests that resulted in a significant difference between groups. We have reported only *p*-values for statistical comparisons involving two-sample Kolmogorov-Smirnov tests as they do not generate any F-statistic. In addition, we have also included the source data sheet provided before, which is comprised of a detailed summary of relevant statistics, including F-values, t-values, and degrees of freedom for two-tailed unpaired t-tests and D-values for two-sample Kolmogorov-Smirnov tests. We have also included the sample size, including number of animals and number of cells as required (also indicated in figure legends within the main manuscript).